# Multifactor Market Indexes

**Wei Liu [1] and James W. Kolari [2],***

1    USAA Bank, San Antonio, TX 78288, USA; yuna0831@gmail.com
2    Department of Finance, Mays Business School, Texas AM University, College Station, TX 77843, USA
*    Correspondence: j-kolari@tamu.edu

**Abstract:** This paper combines the CRSP market index with multiple factors to create a single multi-factor market index. Empirical tests of different multifactor market indexes indicate that: (1) Sharpe ratios substantially increase and GRS test statistics decrease as multifactors are incrementally added to the CRSP index; and (2) the resultant multifactor market indexes are significantly priced in cross-sectional tests of associated beta loadings with *t*-values exceeding 3.0 in most cases.

**Keywords:** market index; market factor; multifactors; efficient portfolios

## 1. Introduction

Cochrane (2011) has humorously alluded to the growing list of multifactors in asset pricing as a "factor zoo". With so many contenders as factors, what factors should be used in an asset pricing model? Harvey et al. (2016) investigated over 300 proposed multifactors in asset pricing models in an effort to discern significant factors versus false discoveries. In addition, Chordia et al. (2020) examined false factor discoveries in asset pricing by studying 2 million trading strategies with real data. Both studies recommended that *t*-statistics associated with factor loadings in Fama and MacBeth (1973) cross-sectional regression tests should exceed 3 to avoid false factor discoveries. This higher statistical hurdle substantially reduces the number of acceptable factors.

Given a smaller set of factors, a multifactor mall problem still remains. For example, assuming 20 valid factors, a large number of models are conceivable. Which combinations of factors and models should be used? In this paper, we propose a solution to this problem. Specifically, we aggregate numerous factors into a single market index. We hypothesize that the CRSP market index (i.e., market portfolio proxy) can be combined with popular multifactors to create more efficient aggregate market indexes dubbed multifactor market indexes. The rationale for these new aggregate indexes is straightforward. If the CRSP index and investable multifactors provide a well-specified asset pricing model for U.S. stock returns, then combining them into a single mean-variance efficient portfolio should be possible. Assuming that a multifactor market index is a linear combination that is efficient, it should be significantly priced as a single factor in a market beta pricing model. Hence, a large number of investable multifactors can be incorporated into a single market index to develop low-dimensional and more parsimonious models. While numerous authors have proven in theory that a combination of factor-mimicking portfolios is minimum variance efficient (e.g., Huberman and Kandel (1987); Shanken (1987); Fama (1996); Shanken and Weinstein (2006); Kan and Zhou (2008), and others), no previous studies to the authors' knowledge have verified this result in empirical tests. Filling this gap in the literature, we incrementally combine popular factors with the CRSP index to show that the resultant aggregate market indexes are increasingly efficient and priced in the cross section of average stock returns.

To construct efficient multifactor market indexes, we utilize portfolio weights for the CRSP index and each multifactor based on their relative Sharpe ratios. In this regard, we incrementally add five prominent multifactors, including size, value profit, investment,

and momentum, to the value-weighted CRSP index. As expected, Sharpe ratios gradually increase as more multifactors are incorporated into the CRSP index. For example, in the sample period July 1963 to December 2016, the CRSP index has a Sharpe ratio of 0.12 compared to 0.34 for a market index combining the CRSP index with the aforementioned five multifactors. GRS time-series regression tests indicate that market indexes become gradually more efficient as multifactors are added. Importantly, cross-sectional Fama and MacBeth regression tests show that multifactor market indexes become increasingly significant as multifactors are incorporated into the CRSP index. We obtain *t*-values associated with multifactor market index beta loadings greater than 3.0 for numerous test asset portfolios, which exceeds the recommended threshold for statistical significance. Hence, multifactors improve approximations of efficient market portfolio returns in the sense that multifactor market indexes become significantly priced in the cross-section of average stock returns.

Further analyses construct an aggregate industry index using our relative Sharpe ratio weighting procedure. It is well known that industry portfolios are not priced by the CRSP index and popular multifactors. By contrast, we find that our new industry index is significantly priced using industry portfolios as test assets. We subsequently combine industry factors and popular multifactors to form a more general multifactor market index that garners high *t*-values typically ranging from 3 to 6 for a variety of test assets, in addition to insignificant mispricing errors and significant pricing of industry portfolios.

Regarding model selection, we find that the momentum factor continues to be significantly priced in cross-sectional tests even when incorporated in multimarket indexes. Hence, we interpret this evidence to mean that momentum is a possible strong factor. Of course, as more multifactors are incorporated into multifactor market indexes to boost their efficiency, the hurdle for strong factors is raised. Future studies may well find that momentum is not a strong factor in the case of more efficient market multifactor indexes. More generally, it is likely that few strong factors exist, such that low-dimensional models are possible.

We conclude that combining multifactors with the CRSP index enables the formation of single market indexes that are efficient. Unlike many studies that reject market beta using the CRSP market index and other stock market indexes, multifactor market indexes lend support for the notion that asset returns are a linear function of a general market index. An important implication is that asset pricing models incorporating a market factor can benefit from multifactor market factors. Currently, the CRSP index continues to be used as the market proxy in most asset pricing models even though it is not priced in the cross section of stock returns. Multifactor market indexes that are significantly priced can potentially lead to more parsimonious and robust asset pricing models. Since the market factor is commonly used in many areas of corporate, investment, and institutional finance, widespread applications of multifactor market indexes are possible in future studies. Another implication is that similar efficient aggregate indexes can be constructed in the real world for investment purposes, thereby benefiting many investors including those saving for retirement.

Section 2 provides background discussion of multifactor market indexes. Section 3 combines the CRSP index with popular multifactors to form a variety of multifactor market indexes based on U.S. stock returns. Descriptive statistics, times-series tests, and cross-sectional tests of multifactor market indexes are provided, in addition to results for an aggregate industry factor and discussion of momentum as a possible strong factor. Section 4 concludes.

## 2. Multifactor Market Indexes

The Capital Asset Pricing Model (CAPM) of Treynor (1961, 1962), Sharpe (1964), Lintner (1965), Mossin (1966), and Black (1972) is based on a mean-variance efficient market portfolio computed as the value-weighted return on all marketable assets. Roll (1977), and others have shown that this portfolio is a minimum variance portfolio if and only if beta

associated with the market factor is priced for all assets in this portfolio. Unfortunately, Fama and French (1992, 1993, 1995, 1996a, 1996b) found that CAPM beta loadings associated with the market portfolio proxied by the CRSP stock index timates were not significantly priced in the cross section of average stock returns. In view of this failure, they concluded that CAPM beta was dead and proposed a three-factor model that augments the CRSP market factor with largely orthogonal size and value factors defined as zero-investment portfolios with long and short positions. The success of this multifactor innovation triggered a plethora of related studies.

Due to the CAPM's failure as discussed in the previous section, multifactor models have arisen with theoretical support from Ross' (1976) arbitrage pricing theory (APT) and Merton's (1973) intertemporal capital asset pricing model (ICAPM). Carhart (1997) added a zero-investment momentum factor to the three-factor model to study mutual fund performance. Subsequently, Chen and Zhang (2010) advanced another three-factor model by replacing size and value with profit and investment factors. A related paper was later published by Hou et al. (2014), which proposed a $q$-factor model with four factors (viz., market, size, investment, and return on equity) grounded in neoclassical investment $q$-theory. Similar to the $q$-factor model, Fama and French (2015) added profit and investment factors to their three-factor model to create a five-factor model. In addition, Stambaugh and Yuan (2017) proposed a four-factor model including the market and size factors plus two mispricing factors (viz., management and performance).[1] Hou et al. (2018) added a growth factor to the $q$-model. Barillas and Shanken (2018) formed a six-factor model by including value and momentum factors. Additionally, Fama and French (2018) added momentum to their five-factor model to form a six-factor model. Subsequently, Fama and French (2020) proposed cross-section factors developed from Fama–MacBeth regressions as well as conditional models with time-varying factor loadings. Many other models containing different factors have been proposed in the asset pricing literature.[2] This proliferation of factors has resulted in a *model mall problem*. Which model should an academic researcher or professional investment manager use? Is there a way to condense well-accepted factors into a more parsimonious model?

Using the notation and discussion in Ferson (1995, 2019), multifactor models of the expected return on the $i$th asset take the familiar cross-sectional form:

$$E_t(R_{i,t+1}) = \lambda_{0t} + \sum_{k=1}^{K} b_{ikt}\lambda_{kt}, \text{for all } i, \tag{1}$$

where $b_{i1t}, \ldots, b_{iKt}$ are time $t$ conditional betas for asset $i$ related to $K$ risk factors, and $\lambda_{kt}$ are market-wide risk premiums for $k = 1, \ldots, K$ risk factors equal to the incremental expected return per unit type-$j$ beta. The intercept $\lambda_{0t}$ is the riskless return or expected zero-beta rate conditionally uncorrelated with the $K$ risk factor loadings if no riskless asset exists. The conditional betas are estimated from the time-series factor model:

$$R_{i,t+1} = a_{it} + \sum_{k=1}^{K} b_{ijt}F_{k,t+1}, \text{for all } i, \tag{2}$$

where $E_t(\mu_{i,t+1}F_{k,t+1}) = E_t(\mu_{i,t+1}) = 0$ for all $i$ and $k$. The model is well-specified when the factor portfolios form the tangency portfolio (i.e., $a_{it} = 0$). As proven by many authors, Equation (2) implies that a combination of $K$ factor-mimicking portfolios is the minimum variance efficient (see Grinblatt and Titman (1987); Huberman et al. (1987); Jobson and Korkie (1982); Gibbons et al. (1989); Kan and Zhou (2008), and Ferson and Siegel (2009), among others).

Extending this literature, MacKinlay (1993) argued that, if a linear combination of factor portfolios cannot identify the efficient tangency portfolio, there exists an optimal orthogonal portfolio of $N$ assets, which when combined with $K$ factor portfolios, forms the tangency portfolio. This unique portfolio is orthogonal to the factor portfolios.[3] Using

this approach, he showed that a bound on the Sharpe ratio exists if the deviation from the single factor model (e.g., CAPM) can be accomplished by a common component of the residual variance. In other words, it is possible that the tangency portfolio can be located by identifying a complete (but limited) set of orthogonal risk factors. This tangency portfolio has the maximum squared Sharpe measure among all portfolios.

In this paper we employ multifactors to develop more efficient aggregate indexes of the stock market. The value-weighted CRSP stock market index is used to proxy the theoretical orthogonal optimal portfolio. We assume that all market information is known and investors seek the most efficient portfolio based on available information. Given the long-only portfolio excess return denoted as $R_L$, we add $k = 1, \ldots, K$ orthogonal factor portfolios to form new more efficient portfolios with higher Sharpe ratios. Factor portfolios are investable zero-investment portfolios.[4] We define the corresponding Sharpe ratios as

$$
\begin{aligned}
S_L &= \frac{\mu_L}{\sigma_L} \\
S_k &= \frac{\mu_k}{\sigma_k}, \; k = 1, \ldots, K,
\end{aligned}
\tag{3}
$$

where $\mu$ is the excess return, and $\sigma$ is the volatility of the portfolio. Combining the long portfolio with the first zero-investment (long/short) factor portfolio $F_1$, the aggregate index return $R_I$ is

$$
R_I = R_L + x_1 F_1,
\tag{4}
$$

with variance equal to $\sigma_L^2 + x_1^2 \sigma_1^2$ and Sharpe ratio

$$
S_I = \frac{\mu_L + x_1 \mu_1}{\sqrt{\sigma_L^2 + x_1^2 \sigma_1^2}}.
\tag{5}
$$

The first order condition $\partial S_I / \partial x_1$ gives

$$
x_1 = \frac{S_1}{S_L} \frac{\sigma_L}{\sigma_1} = \frac{S_1^2}{S_L^2} \frac{\mu_L}{\mu_1}.
\tag{6}
$$

Now the Sharpe ratio for market index $I$ can be written as

$$
S_I = \frac{\mu_L + \left(\frac{S_1^2}{S_L^2} \frac{\mu_L}{\mu_1}\right)\mu_1}{\sqrt{\sigma_L^2 + \left(\frac{S_1}{S_L} \frac{\sigma_L}{\sigma_1}\right)^2 \sigma_1^2}} = \frac{\left(1 + \frac{S_1^2}{S_L^2}\right)\mu_L}{\sqrt{1 + \frac{S_1^2}{S_L^2}} \sigma_L} = \sqrt{1 + \frac{S_1^2}{S_L^2}} S_L > S_L
\tag{7}
$$

Thus, this new market index portfolio is more efficient than the long-only portfolio index. Upon continuing this process by incrementally adding more zero-investment factors to candidate portfolio $P$, the market index portfolio's efficiency is increased. As recognized by MacKinlay (1993), the number of true risk factors will be limited even for increasing numbers of assets in the market. Of course, if a candidate market index portfolio is the tangency portfolio, it is not possible to increase the Sharpe ratio by combining it with another zero-investment factor.[5]

It is worthwhile noting that Equation (7) can alternatively be written as

$$
S_I = \sqrt{S_L^2 + S_I^2}.
\tag{8}
$$

After iteratively adding $k = 1, \ldots, K$ factors, the optimized Sharpe ratio is

$$S_F = \sqrt{\sum_{k=1}^{K} S_k^2} = \sqrt{\mu' \Sigma^{-1} \mu}, \tag{9}$$

which is the ex post tangency portfolio defined by Gibbons et al. (1989). Note that this approach does not provide the weight $x_i$ in Equation (6) to form efficient portfolio indexes.

In the next section we conduct empirical tests of these concepts. First, long/short portfolios based on zero-investment portfolios (or factors) are added to the CRSP index to determine whether more efficient market indexes can be constructed. The market indexes should become increasingly efficient as zero-investment portfolios are added based on size, value, profit, capital investment, and momentum. We also form a market index that incorporates both the aforementioned multifactors plus zero-investment portfolios that are long industry portfolios and short Treasury bills. Second, we perform time-series and cross-sectional tests of multifactor market indexes to determine if they are significantly priced and therefore efficient.

## 3. Empirical Tests

Here we report the empirical results for multifactor market indexes based on the following popular risk factors: market (CRSP index), size (SMB), value (HML), profit (RMW), capital investment (CMA), and momentum (MOM). Further analyses incorporate zero-investment industry portfolios as well. Monthly returns for the value-weighted CRSP market index, zero-investment multifactors, industry portfolios, and Treasury bills are downloaded from Kenneth French's website.[6] Using the weighting procedure in the previous section, we construct multifactor market indexes that combine the CRSP index with five multifactors.

Because factors are not uncorrelated, we orthogonalize them as follows. Denoting the current market index return as $R(I_K)$ (e.g., CRSP index) and the multifactor return (e.g., size factor) to be added to this market index as $F_{k+1}$, we regress $F_{k+1}$ on the excess market return $R(I_{K,t}) - R_{f,t}$ over $t$ sample period months:

$$F_{k+1,t} = \alpha_{k+1} + \beta_{k+1}[R(I_{K,t}) - R_{f,t}] + \epsilon_{k+1,t}. \tag{10}$$

The orthogonalized multifactor is $F_{k+1,t}^{new} = F_{k+1,t} - \beta_{k+1}[R(I_{K,t}) - R_{f,t}]$. The portfolio $I_{K+1}$ return combining the previous portfolio $I_K$ return plus the multifactor portfolio return is:

$$R(I_{K+1,t}) = R(I_{K,t}) + x_{k+1} F_{k+1,t}^{new}. \tag{11}$$

After substituting $F_{k+1,t}^{new}$ and rearranging terms, we have:

$$R(I_{K+1,t}) = R_{f,t} + (1 - x_{k+1}\beta_k)[R(I_{K,t}) - R_{f,t}] + x_{k+1} F_{k+1,t}. \tag{12}$$

This equation decomposes the new multifactor market index return $R(I_{K+1,t})$ into the riskless rate plus weighted premiums for the previous index excess return $R(I_{K,t}) - R_{f,t}$ and newly-added zero-investment factor return $F_{k+1,t}$. Notice that the estimated $\beta_{k+1}$ coefficient in Equation (10) affects the relative weights in the construction of return series $R(I_{K+1,t})$. As noted by an anonymous referee, errors in estimated coefficients will affect these weights and, in turn, multifactor market index returns. This potential bias is beyond the scope of the present research and is therefore left for future research.

To determine the weight $x_k$ for the new factor in this multifactor market index, we apply Equation (6). These steps are repeated to incrementally add the five popular long/short factors $F_{k+1}$ ($k + 1 = 2, \ldots, 6$) to each successive market index $I_{K+1,t}$ in month $t$ to create five new market indexes with respective monthly return series $R(I_{K+1,t})$ (i.e., $R(I_2, t), \ldots, R(I_6, t)$). The return series $R(I_{K,t}) = R(I_1, t)$ is the CRSP index.

### 3.1. Descriptive Statistics

Based on the sample period from July 1963 to December 2016, descriptive statistics in Table 1 are provided for the CRSP index (denoted $I_1$), five multifactors (i.e., SMB, HML, RMW, CMA, and MOM), and multifactor market indexes combining the CRSP index with various multifactors (denoted $I_2$ to $I_6$). The familiar CRSP index ($I_1$) has mean excess returns of 0.51 percent per month, standard deviation of 4.42 percent, and Sharpe ratio of 0.12. With the exception of momentum, the descriptive statistics for the multifactors are not too different from one another. Momentum (MOM) has a noticeably higher mean return of 0.66 percent per month compared to the excess return on the CRSP index, whereas the other multifactors have mean returns less than the excess return on the CRSP index in the range of 0.24 percent to 0.37 percent.

**Table 1.** Descriptive statistics for the CRSP index, five multifactors, and market indexes combining the CRSP index with various mult-factors and industry portfolios: July 1963–December 2016. Based on CRSP stock return data in the sample period July 1963 to December 2016, this table provides descriptive statistics for monthly excess returns (over the Treasury bill rate) on the value-weighted CRSP index, zero-investment portfolio returns of five popular multifactors, and excess returns of seven market indexes combining CRSP index excess returns with these multifactors in addition to industry excess returns. We downloaded the following multifactors and industry returns from Kenneth French's website: size (SMB), value (HML), profit (RMW), capital investment (CMA), momentum (MOM), and 30 industries. Multifactor market indexes are formed using the following steps. First, the size (SMB) factor monthly returns are regressed on CRSP index ($I_1$) excess returns over the Treasury bill rate. The residual term from this regression is utilized as the orthogonalized factor. Second, this orthogonalized size factor is added to the CRSP index using Equation (11) to compute the return for the new multifactor market index $I_2$ as $R(I_2) = R(I_1) + x_1 SMB$. Third, value (HML) factor returns are regressed on the new $R(I_1) + x_1 SMB$ portfolio excess returns to obtain the orthogonalized value factor. This residual value factor is added to the CRSP + SMB portfolio to get market index $I_3 = CRSP + x_1 SMB + x_2 HML$. Fourth, the last step is repeated to sequentially create market index $I_4 = CRSP + x_1 SMB + x_2 HML + x_3 RMW$, market index $I_5 = CRSP + x_1 SMB + x_2 HML + x_3 RMW + x_4 CMA$, and market index $I_6 = CRSP + x_1 SMB + x_2 HML + x_3 RMW + x_4 CMA + x_5 MOM$. We drop the CMA multifactor to form market index $I_7 = CRSP + x_1 SMB + x_2 HML + x_3 RMW + x_5 MOM$. Lastly, 30 industry excess returns are added to create market index $I_8$. For comparison purposes, mean excess returns, standard deviations of returns, and Sharpe ratios for multifactors and different market indexes are computed.

| Portfolios | Mean | Std. Dev. | Sharpe Ratio |
|---|---|---|---|
| $I_1$ = CRSP | 0.51 | 4.42 | 0.12 |
| SMB | 0.27 | 3.04 | 0.09 |
| HML | 0.37 | 2.81 | 0.13 |
| RMW | 0.24 | 2.23 | 0.11 |
| CMA | 0.31 | 2.01 | 0.15 |
| MOM | 0.66 | 4.22 | 0.16 |
| $I_2$ = CRSP + SMB | 0.64 | 4.94 | 0.13 |
| $I_3$ = CRSP + SMB + HML | 1.71 | 8.10 | 0.21 |
| $I_4$ = CRSP + SMB + HML + RMW | 2.66 | 10.09 | 0.26 |
| $I_5$ = CRSP + SMB + HML + RMW + CMA | 2.98 | 10.69 | 0.28 |
| $I_6$ = CRSP + SMB + HML + RMW + CMA + MOM | 4.49 | 13.13 | 0.34 |
| $I_7$ = CRSP + SMB + HML + RMW + MOM | 4.22 | 12.72 | 0.33 |
| $I_8$ = CRSP + SMB + HML + RMW + CMA + MOM + 30 Industry Factors | 6.19 | 15.41 | 0.40 |

Referring to the market index results, the index portfolio denoted $I_2$ = CRSP + SMB in Table 1 has mean excess return, volatility, and Sharpe ratio characteristics similar to momentum. As more multifactors are added to the CRSP index, new market indexes have noticeably higher mean excess returns, volatility, and, more importantly, Sharpe ratios compared to the CRSP index. For example, for the CRSP index plus all five multifactors

(denoted $I_6$ = CRSP + SMB + HML + RMW + CMA + MOM), the mean excess return per month jumps to 4.49 percent, standard deviation of returns to 13.13 percent, and Sharpe ratio to 0.34. The latter Sharpe ratio is almost three times that of the CRSP index. Hence, even though this market index has considerably higher total risk compared to the CRSP index, its excess return per unit risk is much higher than this commonly-used market index. We computed a variety of multifactor market indexes with different combinations of multifactors and find that dropping the CMA multifactor from the market index (i.e., $I_7$ = CRSP + SMB + HML + RMW + MOM) has little effect on the descriptive statistics of the market index (e.g., Tables 1 and 2 show that the Sharpe ratio only decreases from 0.34 to 0.33).

**Table 2.** Descriptive statistics for scaled portfolio indexes combining the CRSP index with multifactors: July 1963–December 2016. This table repeats Table 1 by scaling mean monthly portfolio index returns to contain no leverage. As discussed in the text, to orthogonalize a zero-investment factor (e.g., $F_1$), we regress $F_{1,t}$'s returns on the CRSP index $R(I_{1,t})$ over $t$ sample period months as follows: $F_{1,t} = \alpha_1 + \beta_1 R(I_{1,t}) + \epsilon_{1,t}$. The orthogonalized multifactor is $F_{1,t}^{new} = F_{1,t} - \beta_1 R(I_{1,t})$. Using notation from Equation (11), the more efficient portfolio $I_2$'s return is computed as: $R(I_{2,t}) = R(I_{1,t}) + x_1 F_{1,t}^{new}$. We apply Equation (6) to determine the weight $x_1$ for the factor return $F_{1,t}^{new}$ in the new market index $I_2$. Finally, we divide the mean return $R(I_2)$ over the sample period by the term $1 + x_1(1 - \beta_1)$. Each portfolio index is deleveraged by means of this process.

| Portfolios | Mean | Std. Dev. | Sharpe Ratio |
|---|---|---|---|
| $I_1$ = CRSP | 0.51 | 4.42 | 0.12 |
| SMB | 0.27 | 3.04 | 0.09 |
| HML | 0.37 | 2.81 | 0.13 |
| RMW | 0.24 | 2.23 | 0.11 |
| CMA | 0.31 | 2.01 | 0.15 |
| MOM | 0.66 | 4.22 | 0.16 |
| $I_2$ = CRSP + SMB | 0.40 | 3.07 | 0.13 |
| $I_3$ = CRSP + SMB + HML | 0.38 | 1.82 | 0.21 |
| $I_4$ = CRSP + SMB + HML + RMW | 0.33 | 1.27 | 0.26 |
| $I_5$ = CRSP + SMB + HML + RMW + CMA | 0.33 | 1.18 | 0.28 |
| $I_6$ = CRSP + SMB + HML + RMW + CMA + MOM | 0.38 | 1.11 | 0.34 |
| $I_7$ = CRSP + SMB + HML + RMW + MOM | 0.39 | 1.18 | 0.33 |
| $I_8$ = CRSP + SMB + HML + RMW + CMA + MOM + 30 Industry Factors | 0.40 | 0.98 | 0.40 |

In constructing aggregate market indexes combining the CRSP index with the SMB, HML, RMW, CMA, and MOM multifactors, the weights for factors denoted $x_k$ in Equation (6) for multifactor market indexes are as follows:[7]

$I_2$ = 0.533 CRSP + 0.467 SMB;

$I_3$ = 0.313 CRSP + 0.223 SMB + 0.464 HML;

$I_4$ = 0.231 CRSP + 0.175 SMB + 0.248 HML + 0.346 RMW;

$I_5$ = 0.232 CRSP + 0.175 SMB + 0.073 HML + 0.203 RMW + 0.317 CMA;

$I_6$ = 0.197 CRSP + 0.120 SMB + 0.123 HML + 0.159 RMW + 0.203 CMA + 0.198 MOM;

$I_7$ = 0.202 CRSP + 0.130 SMB + 0.202 HML + 0.236 RMW + 0.230 MOM.

In most of the indexes, SMB gets a relatively lower weight and therefore contributes less to increasing the Sharpe ratio of indexes than other factors. No individual factor appears to dominate the other factors in terms of relative weight. In general, all of the factors are important in forming aggregate indexes.

Our multifactor market indexes have noticeably higher total risk than the CRSP index due to increasing leverage in these portfolios. By deleveraging these indexes, a more accurate assessment of how multifactors affect both mean excess returns and their standard deviation can be obtained. In this respect, Sharpe ratios adjust mean excess returns for total risk but do not reveal these component effects. To adjust for leverage, we rescale the market index portfolios to a zero leverage level. As an example, consider portfolio index

$I_2$ combining the CRSP index with the size factor denoted $F_1$. We initially orthogonalize size factor returns $F_1$. To simplify the derivation and avoid abusing notations, we substitute $R(I_{1,t})$ for $R(I_1, t) - R_{f,t} = R_{m,t} - R_{f,t}$ in Equation (10). Now the orthogonalized multifactor is $F_{1,t}^{new} = F_{1,t} - \beta_1 R(I_{1,t})$. Portfolio index $I_2$'s risk premium is computed as: $R(I_{2,t}) = R(I_{1,t}) + x_1 F_{1,t}^{new} = R(I_{1,t}) + x_1[F_{1,t} - \beta_1 R(I_{1,t})]$. To deleverage the index $I_2$'s return, we divide the mean return $R(I_2)$ over the sample period by the term $1 + x_1(1 - \beta_1)$.[8] This process is repeated for portfolio indexes' returns $R(I_3)$ to $R(I_7)$. In this way, each portfolio index is deleveraged.

Portfolio index returns with no leverage are shown in Table 2. As in Table 1, the highest Sharpe ratios are attained by indexes $I_6$, $I_7$, and $I_8$ at 0.34, 0.33, and 0.40, respectively, which are approximately three times the CRSP index at 0.12. Notice that most of this gain in efficiency is due to decreasing the standard derviation of returns. Indexes $I_2$ to $I_8$ have lower mean returns than the CRSP index but substantially lower standard deviations of returns. Portfolios $I_5$ to $I_8$ have less than one-third of the standard deviation of CRSP index returns. Thus, adding multifactors and industry factors to the CRSP index provides sizable diversification benefits in the form of lower total risk. As multifactors and industry factors are added to the market index, new multifactor market indexes gradually become more efficient with higher Sharpe ratios due to diversification gains.

*3.2. Time-Series Tests of Multifactor Market Indexes*

Stambaugh (1982) created alternative market indexes combining common stocks with bonds, real estate, and consumer durables and found very high correlations between their time-series returns. Not surprisingly, empirical tests of the CAPM were not sensitive to the composition of these market indexes. Consistent with these findings, Black (1995) commented that "...all candidates for the U.S. market portfolio are highly correlated...", including U.S. domestic and world market indexes, equal- and value-weighted portfolios, and human capital and real estate portfolios of traded assets.[9] For this reason, he believed that the problem of selecting an appropriate market index was not severe (even though it tends to flatten the line between expected return and beta). Related work by Jagannathan and Wang (1996) augmented the value-weighted market index with a proxy for human capital to more comprehensively measure the return on aggregate wealth. Unlike Stambaugh, rather than combining returns to stock market capital and human capital to form a single market index, they treated them as two different market factors in a CAPM market model framework. Subsequent cross-sectional tests indicated that human capital was significantly priced but not the value-weighted market index.[10]

In Table 3 we report the correlation coefficients between the time-series monthly returns for different multifactors and market indexes. The multifactors themselves tend to have relatively low correlation coefficients, with the exception of HML and CMA at 0.69. As a multifactor is added to a market index, their correlation naturally increases (e.g., the CRSP index and SMB have a correlation of 0.28 compared to the correlation of market index CRSP + SMB and SMB at 0.68). In addition, as multifactors are progressively added to create new market indexes, the correlation of the CRSP index with market indexes decreases. Strikingly, the correlation between the CRSP index and market index $I_6$ containing the five multifactors is only 0.06. Multifactor market indexes are not only more efficient than the CRSP index but not highly correlated with this index (and other commonly-used market indexes) due to including the net effects from other risk factors with relatively low correlations with the CRSP index.

Gibbons et al. (1989) developed a time-series regression test of the CAPM. The GRS statistic tests estimated whether the $\alpha_i$s in the CAPM market model (1) for $i = 1, \ldots, A$ test assets jointly equal zero. Alternatively, given a set of test assets, a riskfree rate, and a market index, the results can be interpreted as a test of whether the market index is a mean-variance efficient portfolio (see Fama 2017). Using monthly returns for Fama and French's 25 size and book-to-market (value) sorted test asset portfolios downloaded from Kenneth French's website for the sample period July 1963 to December 2016, we estimated

market model (1) with different market indexes. The GRS test statistics in Table 4 indicate that none of the market indexes is a mean-variance efficient portfolio (i.e., all *F*-values are statistically significant at the 1 percent level).[11] However, as multifactors are added to create new market indexes, GRS statistics gradually decrease from 4.77 for the CRSP index to 2.77 for market index $I_7$ = CRSP + SMB + HML + RMW + MOM. We infer that, for these test assets, market indexes become gradually more efficient as multifactors are added. In this regard, more multifactors are needed to achieve an insignificant GRS statistic. Given the large number of factors proposed by researchers as mentioned in the introduction, this possibility is plausible.

**Table 3.** Correlation coefficients between the monthly returns of multifactors and market indexes: July 1963–December 2016. Based on CRSP stock return data in the sample period July 1963 to December 2016, this table provides correlation coefficients between different market indexes as well as multifactor (zero-investment portfolio) returns. Monthly returns for the value-weighted CRSP index and popular multifactors are downloaded from Kenneth French's website. The multifactors are: size (SMB), value (HML), profit (RMW), capital investment (CMA), and momentum (MOM). The market indexes are: $I_1$ = CRSP index, $I_2$ = CRSP + SMB, $I_3$ = CRSP + SMB + HML, $I_4$ = CRSP + SMB + HML + RMW, $I_5$ = CRSP + SMB + HML + RMW + CMA, $I_6$ = CRSP + SMB + HML + RMW + CMA + MOM, $I_7$ = CRSP + SMB + HML + RMW + MOM, and $I_8$ = CRSP + SMB + HML + RMW + CMA + MOM + 30 Industry Factors. The text and Table 1 discuss the process for forming these market indexes that comprise different combinations of CRSP index excess returns, multifactors, and industry excess returns.

| | CRSP | SMB | HML | RMW | CMA | MOM | $I_2$ | $I_3$ | $I_4$ | $I_5$ | $I_6$ | $I_7$ | $I_8$ |
|---|---|---|---|---|---|---|---|---|---|---|---|---|---|
| $I_1$ | 1.00 | 0.28 | −0.26 | −0.23 | −0.38 | −0.13 | 0.90 | 0.51 | 0.33 | 0.15 | 0.06 | 0.21 | 0.27 |
| SMB | | 1.00 | −0.08 | −0.35 | −0.10 | −0.02 | 0.68 | 0.48 | 0.23 | 0.16 | 0.13 | 0.19 | 0.18 |
| HML | | | 1.00 | 0.07 | 0.69 | −0.19 | −0.24 | 0.63 | 0.63 | 0.78 | 0.59 | 0.45 | 0.12 |
| RMW | | | | 1.00 | −0.04 | 0.11 | −0.34 | −0.21 | 0.41 | 0.34 | 0.37 | 0.43 | 0.26 |
| CMA | | | | | 1.00 | −0.01 | −0.34 | 0.29 | 0.25 | 0.55 | 0.49 | 0.21 | 0.38 |
| MOM | | | | | | 1.00 | −0.11 | −0.24 | −0.16 | −0.14 | 0.46 | 0.47 | 0.50 |
| $I_2$ | | | | | | | 1.00 | 0.61 | 0.36 | 0.19 | 0.11 | 0.25 | 0.29 |
| $I_3$ | | | | | | | | 1.00 | 0.80 | 0.79 | 0.57 | 0.56 | 0.33 |
| $I_4$ | | | | | | | | | 1.00 | 0.94 | 0.75 | 0.79 | 0.47 |
| $I_5$ | | | | | | | | | | 1.00 | 0.81 | 0.75 | 0.53 |
| $I_6$ | | | | | | | | | | | 1.00 | 0.96 | 0.77 |
| $I_7$ | | | | | | | | | | | | 1.00 | 0.73 |
| $I_8$ | | | | | | | | | | | | | 1.00 |

**Table 4.** GRS tests using 25 size-value sorted test asset portfolios. This table reports GRS statistic results proposed by Gibbons et al. (1989), which tests whether all estimated intercepts ($\alpha$s) for the test assets jointly equal zero. Our test assets are the monthly returns for the 25 size-value portfolios downloaded from Kenneth French's data website for the sample period July 1963 to December 2016. The GRS test is conducted for the CAPM market model estimated with the following market indexes: $I_1$ = CRSP index, $I_2$ = CRSP + SMB, $I_3$ = CRSP + SMB + HML, $I_4$ = CRSP + SMB + HML + RMW, $I_5$ = CRSP + SMB + HML + RMW + CMA, $I_6$ = CRSP + SMB + HML + RMW + CMA + MOM, $I_7$ = CRSP + SMB + HML + RMW + MOM, and $I_8$ = CRSP + SMB + HML + RMW + CMA + MOM + 30 Industry Factors. The text and Table 1 discuss the process for forming these market indexes that represent different combinations of CRSP index excess returns, multifactors, and industry excess returns. The GRS statistic follows an $F(25, 642 − 25 − 1)$ distribution. Greater GRS test values indicate larger absolute values of estimated $\alpha$s.

| | $I_1$ = CRSP | $I_2$ | $I_3$ | $I_4$ | $I_5$ | $I_6$ | $I_7$ | $I_8$ |
|---|---|---|---|---|---|---|---|---|
| *F*-value | 4.78 | 4.70 | 3.85 | 3.38 | 3.55 | 2.86 | 2.77 | 2.81 |

### 3.3. Cross-Sectional Fama-MacBeth Tests of Multifactor Market Indexes

An important test of the efficiency of market indexes augmented with multifactors is whether they are significantly priced in the cross section of average stock returns. For this purpose, using monthly excess returns for 25 size-value sorted test asset portfolios downloaded from French's website, we conduct Fama and MacBeth (1973) tests of market beta associated with different market indexes incorporating popular multifactors. Tests incorporating industry factors are provided in the forthcoming Section 3.6.

We begin by estimating time-series regressions for the full sample period using the excess returns for each of the 25 portfolios and 1 of the market indexes. Monthly index returns are scaled to contain no leverage (see Table 2). As such, the estimated index premiums can be compared to the historical market premium (i.e., 0.51 percent per month in Table 1). Rolling monthly cross-sectional regressions are estimated with excess returns in month $t$ and the estimated full sample betas for all sample period months $t = 1, \ldots, T$. From these regressions we estimate the market price of beta risk, or $\hat{\lambda}_{Mt}$, which are averaged over $t = 1, \ldots, T$ sample months to obtain $\hat{\lambda}_M$. This period-by-period regression approach has the advantage that the $t$-statistic associated with $\hat{\lambda}_M$ takes into account the covariance of regression residuals and the independent variables without requiring estimates of the covariances (see Fama 2017). The analyses are repeated for each of the portfolio market indexes.

Results for the cross-sectional tests are reported in Table 5. Consistent with earlier studies, the CRSP index ($I_1$) is not significantly priced, as the market price of risk $\hat{\lambda}_M = -0.43$ ($t = -1.06$). Additionally, index $I_2$ = CRSP + SMB produces a market proxy that is not priced with $\hat{\lambda}_M = 0.00$ ($t = 0.00$). However, upon adding both SMB and HML to the CRSP index to form index $I_3$, we obtain $\hat{\lambda}_M = 0.34$ ($t = 3.80$), which is economically meaningful and statistically significant at less than a one percent level. Further adding RMA, CMA, and MOM multifactors to the CRSP index to form index $I_6$ yields similar results. Notice that the strongest finding is for multifactor market index $I_7$ = CSRP + SMB + HML + RMW + MOM (excluding CMA), which yields $\hat{\lambda}_M = 0.38$ ($t = 4.04$). In addition, the estimated mispricing term $\hat{\alpha}$ is insignificantly different from zero for the following market indexes: $I_3$ = CRSP + SMB + HML, $I_4$ = CRSP + SMB + HML + RMW, and $I_7$ = CRSP + SMB + HML + RMW + MOM. Finally, adjusted $R^2$ values jump from 9 percent for the CRSP index to over 50 percent for the combined CRSP plus multifactor portfolios, with the exception of the CRSP + SMB portfolio.[12] For market index $I_7$ = CRSP + SMB + HML + RMW + MOM, the adjusted $R^2$ value is 0.71, which indicates a fairly strong goodness-of-fit.

Importantly, the $t$-values associated with a number of multifactor market indexes well exceed recent standards for the significance of an asset pricing factor. As mentioned earlier, Harvey et al. (2016) and Chordia et al. (2020) have recommended that acceptable factors should exceed a $t$-statistic threshold of 3.0 or more. We infer that market indexes combining the CRSP index and multifactor portfolios provide market portfolio proxies that are priced in the cross-section of average stock returns and therefore are relatively more efficient portfolios than the CRSP index.

### 3.4. Cross-Sectional Fama-MacBeth Tests of Multifactors

Here we compare the cross-sectional test results for the CAPM, three-factor, and five-factor models, in addition to the five-factor model augmented with the momentum factor. The results in Table 6 show that the five-factor model plus momentum has insignificant mispricing (i.e., $\hat{\alpha} = 0$) and a higher estimated adjusted $R^2$ value (i.e., 83 percent) than the other models. Momentum has a $t$-value of 4.71, which exceeds the $t$-values of other factors. Across different models, the SMB, HML, and RMW factors are consistently significant and exceed 3 in some instances, CMA is not priced, and the multifactor models substantially boost the estimated adjusted $R^2$ values relative tthe CAPM. Note that our multifactor market index $I_7$ (see Table 5) outperforms the three- and five-factor models, with similar $R^2$ value but insignificant $\hat{\alpha}$ mispricing, and performs almost as well as the five-factor model plus momentum.

**Table 5.** Cross-sectional asset pricing tests of market beta for different market indexes using 25 size-value sorted test asset portfolios: July 1963–December 2016. Based on CRSP stock return data in the sample period July 1963 to December 2016, this table provides Fama and MacBeth (1973) cross-sectional tests of the value-weighted CRSP index as well as market indexes combining the CRSP index with popular multifactors. Downloaded from Kenneth French's website, the multifactors are: size (SMB), value (HML), profit (RMW), capital investment (CMA), and momentum (MOM). Multifactor market indexes are formed using the following steps. First, the size (SMB) factor monthly returns are regressed on CRSP index ($I_1$) excess returns over the Treasury bill rate. The residual term from this regression is utilized as the orthogonalized factor. Second, this orthogonalized size factor is added to the CRSP index using Equation (11) to compute the return for the new multifactor market index $I_2$ as $R(I_2) = R(I_1) + x_1$SMB. Third, value (HML) factor returns are regressed on the new $R(I_1) + x_1$SMB portfolio excess returns to obtain the orthogonalized value factor. This residual value factor is added to the CRSP + SMB portfolio to get market index $I_3$ = CRSP + $x_1$SMB + $x_2$HML. Fourth, the last step is repeated to sequentially create market index $I_4$ = CRSP + $x_1$SMB + $x_2$HML + $x_3$RMW, market index $I_5$ = CRSP + $x_1$SMB + $x_2$HML + $x_3$RMW + $x_4$CMA, and market index $I_6$ = CRSP + $x_1$SMB + $x_2$HML + $x_3$RMW + $x_4$CMA + $x_5$MOM. In addition, we drop the CMA multifactor to form market index $I_7$ = CRSP + $x_1$SMB + $x_2$HML + $x_3$RMW + $x_5$MOM. Monthly returns for 25 size-value test assets are downloaded from French's data website (i.e., the value firm characteristic corresponds to the book-to-market ratio). A time-series regression using monthly excess returns is run for the full sample period to estimate CAPM betas for each of the 25 tests assets using the CRSP index. Time-series regression analyses are repeated for the other market indexes. Following the procedure in the text and Table 2, monthly market index returns are scaled to contain no leverage. Cross-sectional tests are conducted by estimating monthly rolling cross-sectional regressions with excess returns and full sample betas for all sample months. The resultant monthly series of estimated market prices of beta risk are averaged over all sample months to estimate $\hat{\lambda}_M$ (and associated *t*-statistics are in parentheses).

| Indexes | $\hat{\alpha}$ | $\hat{\lambda}_M$ | Adj. $R^2$ |
|---|---|---|---|
| $I_1$ = CRSP | 1.20 | −0.43 | 0.09 |
| | (3.19) | (−1.06) | |
| $I_2$ = CRSP + SMB | 0.73 | 0.00 | 0.00 |
| | (2.970 | (0.00) | |
| $I_3$ = CRSP + SMB + HML | 0.08 | 0.34 | 0.54 |
| | (0.34) | (3.80) | |
| $I_4$ = CRSP + SMB + HML + RMW | 0.22 | 0.28 | 0.61 |
| | (0.86) | (3.70) | |
| $I_5$ = CRSP + SMB + HML + RMW + CMA | 0.48 | 0.21 | 0.56 |
| | (2.06) | (3.27) | |
| $I_6$ = CRSP + SMB + HML + RMW + CMA + MOM | 0.54 | 0.26 | 0.61 |
| | (2.44) | (3.43) | |
| $I_7$ = CRSP + SMB + HML + RMW + MOM | 0.24 | 0.38 | 0.71 |
| | (0.95) | (4.04) | |

We next consider whether a multifactor is priced when the market index contains the multifactor portfolio. First, we create residual returns for the 25 size-value portfolios (denoted $R_{p,t}^{res}$) as follows:

$$R_{p,t}^{res} = R_{p,t} - R_{f,t} - \beta_p[R(I_{K,t}) - R_{f,t}], \tag{13}$$

where $R_{p,t} - R_{f,t}$ is the excess portfolio return in month $t$, and $R(I_{K,t}) - R_{f,t}$ is the excess market index return as defined in Equation (11), and $\beta_p$ is the estimated beta coefficient for portfolio $p = 1, \ldots, 25$. We employ market indexes $I_1$ to $I_7$ defined earlier in this section. Second, using monthly residual returns and multifactor returns for the full sample period, we run time-series regressions of residual returns on the multifactor returns for SMB, HML, RMW, CMA, and MOM for each of the 25 portfolios. Third, and last, using monthly average

portfolios' residual returns, monthly rolling cross-sectional regressions are run as before to estimate the market prices of risk for the multifactors.

**Table 6.** Cross-sectional asset pricing tests of the CRSP market factor and five popular multifactors: July 1963–December 2016. Based on CRSP stock return data in the sample period July 1963 to December 2016, this table provides Fama and MacBeth (1973) cross-sectional tests of the market factor plus five popular multifactors. Downloaded from Kenneth French's website, the factors are: CRSP market index (M), size (SMB), value (HML), profit (RMW), capital investment (CMA), and momentum (MOM). Test assets are the Fama–French 25 size-value portfolios downloaded from French's data website. For the full sample period, a time-series regression is run using monthly excess returns for test assets as the dependent variable and the CRSP value-weighted market index excess return plus different multifactor returns as the independent variables. Using estimated factor loadings for each of the 25 test asset portfolios, cross-sectional tests are conducted by estimating monthly rolling cross-sectional regressions with monthly excess returns and full sample betas for all sample months. Resultant monthly series of estimated prices of beta risk for the $k$th factor ($k = 1, \ldots, 6$) are averaged over all sample months to estimate $\hat{\lambda}_k$ (and associated $t$-statistics are in parentheses).

| Index | $\hat{\alpha}$ | $\hat{\lambda}_M$ | $\hat{\lambda}_{SMB}$ | $\hat{\lambda}_{HML}$ | $\hat{\lambda}_{RMW}$ | $\hat{\lambda}_{CMA}$ | $\hat{\lambda}_{MOM}$ | Adj. $R^2$ |
|---|---|---|---|---|---|---|---|---|
| CAPM | 1.20 | $-0.43$ | | | | | | 0.09 |
| | (3.19) | $(-1.06)$ | | | | | | |
| Three-factor | 1.27 | $-0.73$ | 0.22 | 0.40 | | | | 0.67 |
| | (4.83) | $(-2.31)$ | (1.78) | (3.50) | | | | |
| Five-factor | 1.02 | $-0.53$ | 0.30 | 0.36 | 0.48 | $-0.02$ | | 0.74 |
| | (3.56) | $(-1.58)$ | (2.47) | (3.15) | (2.82) | $(-0.10)$ | | |
| Five-factor + MOM | 0.28 | 0.26 | 0.33 | 0.39 | 0.61 | $-0.14$ | 2.94 | 0.83 |
| | (0.82) | (0.69) | (2.68) | (3.47) | (3.44) | $(-0.80)$ | (4.71) | |

The empirical results in Table 7 indicate that, with the exception of momentum (MOM), multifactors are not normally priced in the cross-section when the market index contains the respective multifactors. For example, for market indexes $I_1$ and $I_2$ comprised of the CRSP index and CRSP + SML, respectively, HML is significantly priced with $\hat{\lambda}_{HML} = 0.50$ ($t = 4.91$) and 0.61 ($t = 4.93$). However, for market index $I_3$ (viz., CRSP + SMB + HML) containing the HML portfolio, the multifactor HML is not priced with $\hat{\lambda}_{HML} = -0.25$ ($t = -1.13$). RMW is priced using market indexes $I_1$, $I_2$, $I_3$, and $I_6$ but not for $I_4$ and $I_5$ containing the RMW portfolio.[13] Unlike the other multifactors, MOM consistently remains significantly priced across all market indexes, even when the MOM portfolio is included in market indexes $I_6$ and $I_7$. For this reason, we will refer to MOM as a possible strong factor in forthcoming analyses of two-factor models comprised of multifactor market indexes and the momentum factor. From these results we infer that, with the exception of MOM, multifactors are not generally priced when they are incorporated in the market index. Lastly, as multifactors are added to the market index, the explanatory power of the multifactors diminishes (e.g., from 90 percent for the CRSP market index $I_1$ to only 42 percent for the multifactor market index $I_7$ containing SMB, HML, RMW, and MOM).[14]

**Table 7.** Cross-sectional asset pricing tests of multifactors using the excess residual returns of 25 size-value sorted test assets with respect to market index excess returns: July 1963–December 2016. Based on CRSP stock return data in the sample period July 1963 to December 2016, this table provides Fama and MacBeth (1973) cross-sectional tests of five popular multifactors. Downloaded from Kenneth French's website, the multifactors are: size (SMB), value (HML), profit (RMW), capital investment (CMA), and momentum (MOM). Test assets are monthly excess returns (denoted $R_{i,t} - R_{f,t}$) for Fama–French 25 size-value portfolios downloaded from French's data website. The residuals for test assets are computed as $R_{i,t}^{res} = R_{i,t} - R_{f,t} - \hat{\beta}_i R(I_{K,t})$, $i = 1, \ldots, 25$ where $R(I_{K,t})$ is the monthly excess return for one of seven different market indexes ($K = 1, \ldots, 7$). The text and Table 1 discuss the process for forming market indexes that represent different combinations of the CRSP index and multifactors. We denote these market indexes as follows: $I_1 =$ CRSP index, $I_2 =$ CRSP + SMB, $I_3 =$ CRSP + SMB + HML, $I_4 =$ CRSP + SMB + HML + RMW, $I_5 =$ CRSP + SMB + HML + RMW + CMA, and $I_6 =$ CRSP + SMB + HML + RMW + CMA + MOM, in addition to $I_7 =$ CRSP + SMB + HML + RMW + MOM. For the full sample period, a time-series regression is run using monthly residual returns as the dependent variable and multifactor returns as the independent variables to estimate multifactor betas for each of the 25 test asset portfolios. Cross-sectional tests are conducted by estimating monthly rolling cross-sectional regressions with monthly residual returns and full sample betas for all sample months. The resultant monthly series of estimated multifactor prices of beta risk for the $k$th factor ($k = 1, \ldots, 5$) are averaged over all sample months to estimate $\hat{\lambda}_k$ (and associated $t$-statistics are in parentheses).

| Index | $\hat{\alpha}$ | $\hat{\lambda}_{SMB}$ | $\hat{\lambda}_{HML}$ | $\hat{\lambda}_{RMW}$ | $\hat{\lambda}_{CMA}$ | $\hat{\lambda}_{MOM}$ | Adj. $R^2$ |
|---|---|---|---|---|---|---|---|
| $I_1$ | 0.03 | 0.11 | 0.59 | 0.76 | 0.08 | 3.42 | 0.90 |
| | (1.21) | (0.83) | (4.91) | (4.30) | (0.38) | (6.01) | |
| $I_2$ | 0.09 | −0.33 | 0.61 | 0.87 | 0.09 | 3.34 | 0.91 |
| | (5.39) | (−1.59) | (4.93) | (4.88) | (0.40) | (5.98) | |
| $I_3$ | 0.15 | −0.20 | −0.25 | 0.79 | −0.35 | 3.50 | 0.72 |
| | (2.81) | (−1.02) | (−1.13) | (4.37) | (−2.17) | (6.11) | |
| $I_4$ | 0.63 | 0.22 | 0.17 | 0.38 | −0.12 | 2.44 | 0.66 |
| | (3.60) | (1.31) | (0.57) | (1.70) | (−0.86) | (4.18) | |
| $I_5$ | 0.24 | 0.14 | −0.47 | 0.32 | −0.58 | 3.23 | 0.80 |
| | (1.39) | (0.97) | (−1.45) | (1.38) | (−2.74) | (5.48) | |
| $I_6$ | 0.83 | 0.35 | 0.57 | 0.62 | −0.01 | 1.51 | 0.70 |
| | (3.63) | (2.40) | (1.69) | (2.63) | (−0.03) | (2.63) | |
| $I_7$ | 0.90 | 0.37 | 0.53 | 0.58 | 0.03 | 1.25 | 0.42 |
| | (3.94) | (2.27) | (2.10) | (2.37) | (0.17) | (2.34) | |

*3.5. Robustness Tests with Different Test Asset Portfolios*

As a robustness check, we repeat the cross-sectional tests in Table 8 using 25 size-value sorted portfolios for a variety of different test asset portfolios available on French's data website. The following test assets are used: 25 value-investment portfolios, 25 profit-investment portfolios, 25 size-investment portfolios, 25 size-profit portfolios, 32 size-value-investment portfolios, 32 size-value-profit portfolios, 32 size-profit-investment portfolios, and 30 industry portfolios. In general, the results for these portfolios corroborate our findings in Table 5. The $t$-values associated with market indexes $I_4$ to $I_7$ have ranges as follows: 2.32 to 2.52 for 25 value-investment portfolios in Panel A; 3.34 to 3.61 for 25 profit-investment portfolios in Panel B; 4.16 to 4.54 for 25 size-investment portfolios in panel C; 3.85 to 3.89 for 25 size-profit portfolios in Panel D; 3.33 to 3.61 for 32 size-value-investment portfolios in Panel E; 4.49 to 4.91 for 32 size-value-profit portfolios in Panel F; 5.85 to 6.53 for 32 size-profit-investment portfolios in Panel G; and −0.01 to 0.59 for 30 industry portfolios in Panel H. Highlighting the results for the size-profit-investment portfolios in Panel G, the results are as follows: $\hat{\lambda}_M = 0.45$ ($t = 5.85$) for $I_4$, $\hat{\lambda}_M = 0.37$ ($t = 6.20$) for $I_5$, $\hat{\lambda}_M = 0.42$ ($t = 6.53$) for $I_6$, and $\hat{\lambda}_M = 0.51$ ($t = 6.36$) for $I_7$. Mispricing terms $\alpha$ are insignificant for these multifactor market indexes with the exception of $I_6$. Adjusted $R^2$ values range from 0.67 to 0.76. Together, these results corroborate our earlier findings in Table 5 using

size-value test asset portfolios. In addition, the results are comparable to the the three-factor, five-factor, and five-factor plus momentum models in Table 6.

**Table 8.** Robustness checks for cross-sectional asset pricing tests of market beta for different market indexes using a variety of test asset portfolios: July 1963–December 2016. Based on CRSP stock return data in the sample period July 1963 to December 2016, this table provides Fama and MacBeth (1973) cross-sectional tests of the value-weighted CRSP index as well as market indexes combining the CRSP index with popular multifactors. Downloaded from Kenneth French's website, the multifactors are: size (SMB), value (HML), profit (RMW), capital investment (CMA), and momentum (MOM). Multifactor market indexes are formed using the following steps. First, the size (SMB) factor monthly returns are regressed on CRSP index ($I_1$) excess returns over the Treasury bill rate. The residual term from this regression is utilized as the orthogonalized factor. Second, this orthogonalized size factor is added to the CRSP index using Equation (11) to compute the return for the new multifactor market index $I_2$ as $R(I_2) = R(I_1) + x_1$SMB. Third, value (HML) factor returns are regressed on the new $R(I_1) + x_1$SMB portfolio excess returns to obtain the orthogonalized value factor. This residual value factor is added to the CRSP + SMB portfolio to get market index $I_3$ = CRSP + $x_1$SMB + $x_2$HML. Fourth, the last step is repeated to sequentially create market index $I_4$ = CRSP + $x_1$SMB + $x_2$HML + $x_3$RMW, market index $I_5$ = CRSP + $x_1$SMB + $x_2$HML + $x_3$RMW + $x_4$CMA, and market index $I_6$ = CRSP + $x_1$SMB + $x_2$HML + $x_3$RMW + $x_4$CMA + $x_5$MOM. In addition, we drop the CMA multifactor to form market index $I_7$ = CRSP + $x_1$SMB + $x_2$HML + $x_3$RMW + $x_5$MOM. Following the procedure in the text and Table 2, monthly market index returns are scaled to contain no leverage. Monthly returns for a variety of test assets are downloaded from French's data website: 25 value-investment portfolios, 25 profit-investment portfolios, 25 size-investment portfolios, 25 size-profit portfolios, 32 size-value-investment portfolios, 32 size-value-profit portfolios, 32 size-profit-investment portfolios, and 30 industry portfolios (see French's website for details of these portfolios). A time-series regression is run using monthly excess returns for the CRSP index for the full sample period to estimate CAPM betas for each of the tests assets. Time-series regression analyses are repeated for the other market indexes. Cross-sectional tests are conducted by estimating monthly rolling cross-sectional regressions with excess returns and full sample betas for all sample months. The resultant monthly series of estimated market prices of beta risk are averaged over all sample months to estimate $\hat{\lambda}_M$ (and associated *t*-statistics are in parentheses).

**Panel A: 25 Value-Investment Portfolios**

| Indexes | $\hat{\alpha}$ | *t*-Value | $\hat{\lambda}_M$ | *t*-Value | Adj. $R^2$ |
|---|---|---|---|---|---|
| $I_1$ = CRSP | 0.77 | 2.93 | −0.11 | −0.34 | −0.04 |
| $I_2$ = CRSP + SMB | 0.49 | 2.41 | 0.13 | 0.75 | −0.01 |
| $I_3$ = CRSP + SMB + HML | 0.32 | 1.60 | 0.21 | 2.40 | 0.51 |
| $I_4$ = CRSP + SMB + HML + RMW | 0.33 | 1.60 | 0.19 | 2.32 | 0.48 |
| $I_5$ = CRSP + SMB + HML + RMW + CMA | 0.45 | 2.40 | 0.16 | 2.47 | 0.54 |
| $I_6$ = CRSP + SMB + HML + RMW + CMA + MOM | 0.49 | 2.60 | 0.21 | 2.52 | 0.53 |
| $I_7$ = CRSP + SMB + HML + RMW + MOM | 0.30 | 1.38 | 0.29 | 2.39 | 0.50 |

**Panel B: 25 Profit-Investment Portfolios**

| Indexes | $\hat{\alpha}$ | *t*-Value | $\hat{\lambda}_M$ | *t*-Value | Adj. $R^2$ |
|---|---|---|---|---|---|
| $I_1$ = CRSP | 1.29 | 4.91 | −0.69 | −2.23 | 0.25 |
| $I_2$ = CRSP + SMB | 1.05 | 5.13 | −0.35 | −1.95 | 0.19 |
| $I_3$ = CRSP + SMB + HML | 0.45 | 2.10 | 0.09 | 0.74 | −0.02 |
| $I_4$ = CRSP + SMB + HML + RMW | 0.00 | 0.01 | 0.43 | 3.61 | 0.52 |
| $I_5$ = CRSP + SMB + HML + RMW + CMA | 0.36 | 1.70 | 0.26 | 3.36 | 0.54 |
| $I_6$ = CRSP + SMB + HML + RMW + CMA + MOM | 0.46 | 2.32 | 0.29 | 3.34 | 0.59 |
| $I_7$ = CRSP + SMB + HML + RMW + MOM | 0.17 | 0.65 | 0.44 | 3.40 | 0.56 |

**Table 8.** *Cont.*

**Panel C: 25 Size-Investment Portfolios**

| Indexes | $\hat{\alpha}$ | *t*-Value | $\hat{\lambda}_M$ | *t*-Value | Adj. $R^2$ |
|---|---|---|---|---|---|
| $I_1$ = CRSP | 1.04 | 3.83 | −0.28 | −0.84 | 0.01 |
| $I_2$ = CRSP + SMB | 0.62 | 2.94 | 0.07 | 0.43 | −0.03 |
| $I_3$ = CRSP + SMB + HML | 0.20 | 0.94 | 0.30 | 2.32 | 0.39 |
| $I_4$ = CRSP + SMB + HML + RMW | 0.07 | 0.31 | 0.41 | 4.16 | 0.60 |
| $I_5$ = CRSP + SMB + HML + RMW + CMA | 0.42 | 1.90 | 0.30 | 4.52 | 0.62 |
| $I_6$ = CRSP + SMB + HML + RMW + CMA + MOM | 0.53 | 2.43 | 0.34 | 4.54 | 0.64 |
| $I_7$ = CRSP + SMB + HML + RMW + MOM | 0.16 | 0.70 | 0.48 | 4.35 | 0.63 |

**Panel D: 25 Size-Profit Portfolios**

| Indexes | $\hat{\alpha}$ | *t*-Value | $\hat{\lambda}_M$ | *t*-Value | Adj. $R^2$ |
|---|---|---|---|---|---|
| $I_1$ = CRSP | 0.47 | 1.41 | 0.21 | 0.55 | −0.02 |
| $I_2$ = CRSP + SMB | 0.31 | 1.37 | 0.24 | 1.40 | 0.15 |
| $I_3$ = CRSP + SMB + HML | 0.18 | 0.83 | 0.29 | 2.12 | 0.39 |
| $I_4$ = CRSP + SMB + HML + RMW | 0.19 | 0.91 | 0.30 | 3.86 | 0.85 |
| $I_5$ = CRSP + SMB + HML + RMW + CMA | 0.39 | 1.87 | 0.30 | 3.85 | 0.75 |
| $I_6$ = CRSP + SMB + HML + RMW + CMA + MOM | 0.52 | 2.54 | 0.31 | 3.85 | 0.75 |
| $I_7$ = CRSP + SMB + HML + RMW + MOM | 0.32 | 1.57 | 0.32 | 3.89 | 0.84 |

**Panel E: 32 Size-Value-Investment Portfolios**

| Indexes | $\hat{\alpha}$ | *t*-Value | $\hat{\lambda}_M$ | *t*-Value | Adj. $R^2$ |
|---|---|---|---|---|---|
| $I_1$ = CRSP | 0.51 | 1.53 | 0.22 | 0.56 | −0.01 |
| $I_2$ = CRSP + SMB | 0.33 | 1.49 | 0.27 | 1.63 | 0.19 |
| $I_3$ = CRSP + SMB + HML | 0.22 | 1.11 | 0.29 | 3.06 | 0.58 |
| $I_4$ = CRSP + SMB + HML + RMW | 0.29 | 1.39 | 0.25 | 3.33 | 0.46 |
| $I_5$ = CRSP + SMB + HML + RMW + CMA | 0.48 | 2.40 | 0.20 | 3.40 | 0.44 |
| $I_6$ = CRSP + SMB + HML + RMW + CMA + MOM | 0.52 | 2.64 | 0.27 | 3.61 | 0.48 |
| $I_7$ = CRSP + SMB + HML + RMW + MOM | 0.26 | 1.27 | 0.36 | 3.56 | 0.54 |

**Panel F: 32 Size-Value-Profit Portfolios**

| Indexes | $\hat{\alpha}$ | *t*-Value | $\hat{\lambda}_M$ | *t*-Value | Adj. $R^2$ |
|---|---|---|---|---|---|
| $I_1$ = CRSP | 1.17 | 3.54 | −0.42 | −1.15 | 0.02 |
| $I_2$ = CRSP + SMB | 0.60 | 2.71 | 0.08 | 0.48 | −0.02 |
| $I_3$ = CRSP + SMB + HML | −0.01 | −0.05 | 0.40 | 4.23 | 0.56 |
| $I_4$ = CRSP + SMB + HML + RMW | 0.13 | 0.58 | 0.32 | 4.79 | 0.73 |
| $I_5$ = CRSP + SMB + HML + RMW + CMA | 0.40 | 1.87 | 0.26 | 4.49 | 0.66 |
| $I_6$ = CRSP + SMB + HML + RMW + CMA + MOM | 0.49 | 2.38 | 0.31 | 4.54 | 0.67 |
| $I_7$ = CRSP + SMB + HML + RMW + MOM | 0.21 | 0.92 | 0.40 | 4.91 | 0.75 |

**Panel G: 32 Size-Profit-Investment Portfolios**

| Indexes | $\hat{\alpha}$ | *t*-Value | $\hat{\lambda}_M$ | *t*-Value | Adj. $R^2$ |
|---|---|---|---|---|---|
| $I_1$ = CRSP | 1.04 | 3.62 | −0.33 | −0.94 | 0.00 |
| $I_2$ = CRSP + SMB | 0.57 | 2.78 | 0.08 | 0.50 | −0.02 |
| $I_3$ = CRSP + SMB + HML | 0.15 | 0.75 | 0.32 | 2.59 | 0.26 |
| $I_4$ = CRSP + SMB + HML + RMW | −0.03 | −0.12 | 0.45 | 5.85 | 0.67 |
| $I_5$ = CRSP + SMB + HML + RMW + CMA | 0.30 | 1.45 | 0.37 | 6.20 | 0.69 |
| $I_6$ = CRSP + SMB + HML + RMW + CMA + MOM | 0.43 | 2.16 | 0.42 | 6.53 | 0.75 |
| $I_7$ = CRSP + SMB + HML + RMW + MOM | 0.10 | 0.50 | 0.51 | 6.36 | 0.76 |

**Panel H: 30 Industry Portfolios**

| Indexes | $\hat{\alpha}$ | *t*-Value | $\hat{\lambda}_M$ | *t*-Value | Adj. $R^2$ |
|---|---|---|---|---|---|
| $I_1$ = CRSP | 0.67 | 2.99 | −0.06 | −0.20 | −0.03 |
| $I_2$ = CRSP + SMB | 0.66 | 3.64 | −0.03 | −0.18 | −0.03 |
| $I_3$ = CRSP + SMB + HML | 0.71 | 4.24 | −0.06 | −0.52 | 0.01 |
| $I_4$ = CRSP + SMB + HML + RMW | 0.60 | 3.42 | 0.01 | 0.11 | −0.03 |
| $I_5$ = CRSP + SMB + HML + RMW + CMA | 0.61 | 3.31 | 0.00 | −0.01 | −0.04 |
| $I_6$ = CRSP + SMB + HML + RMW + CMA + MOM | 0.60 | 3.04 | 0.03 | 0.35 | −0.02 |
| $I_7$ = CRSP + SMB + HML + RMW + MOM | 0.55 | 2.88 | 0.05 | 0.59 | −0.01 |

Not surprisingly, in Panel H's results for industry portfolios, market indexes $I_4$ to $I_7$ are not significantly priced. It is well-known that common market indexes and popular multi-factors are not priced using exogenous industry portfolios. Hence, a shortfall in the asset priciing literature is the absence of an aggregate industry factor.

### 3.6. Index Construction with Industry Factors

Our findings above indicate that efficient market indexes can be constructed from the CRSP index in combination with popular multifactors. These market indexes do a good job of pricing widely-used test assets with characteristics similar to the multifactors, but are not useful in pricing industry portfolios. Unfortunately, there is no single industry factor available.

Industry returns are different from other common risk factors based on firm characteristics (e.g., size, book-to-market, profit, and capital investment) or stock characteristics (e.g., winner and loser stock returns). A problem in forming an aggregate industry index is that different industries tend to be independent of one another.[15] A shared common risk factor across industries (other than the market factor) is not directly observed. We propose a possible solution to this problem. Using the 30 industry test assets, we initially form a pure long/short portfolio that is long a particular industry and short an equal quantity of Treasury bills. We then combine these 30 industry factors to construct a single industry index based on our multifactor market index methods using relative Sharpe ratio weights in Section 2. In addition, we combine the CRSP index, multifactors, and industry factors to construct a new multifactor market index.

Regarding our earlier cross-sectional tests, we found that the results for a multifactor market index were independent of the order with which multifactors were added one-by-one to the obtained more efficient index. This robustness holds as long as the multifactors themselves are priced factors. Each time we add a new factor to the base market index, we only add in the net part which is not included in the base market index obtained in previous step. It is important to note that, if the new factor has large noise, it will not contribute to increasing the efficiency of the market index. Its noise component will be treated as the net part to add to the base market index and, subsequently, will remain embedded in the market index through the iterative steps of adding different multifactors to the index. To mitigate this potential noise, we add industry factors with relatively smaller volatility prior to higher volatility industry factors. Following this ordered procedure, we compute a single industry index denoted IND.

Cross-sectional asset pricing tests of industry index IND are provided in Table 9. As shown there, when using the 30 industry portfolios as the test assets, this industry index is positively priced, i.e., $\hat{\lambda}_{IND} = 0.82$ ($t = 2.50$).[16] For other test assets based on firm characteristics, however, *IND* is normally significantly priced but negatively so and not priced for profit-investment and size-profit portfolios.

For comparison purposes, we also construct a simple industry index denoted *SIND*, which is defined as an equal-weighted portfolio of the 30 industry factors. The cross-sectional test results in Table 10 show that *SIND* is not priced in the 30 industry portfolios and most other test assets, with the exceptions of being negatively priced for value-investment and profit-investment portfolios.

As a last step, we add the five popular multifactors plus 30 industry portfolios to the CRSP index using relative Sharpe ratio weights as defined in Section 2. Factors are added in the order discussed above, i.e., smaller volatility factors before other factors. The obtained multifactor market index denoted $I_8$ yields the cross-sectional test results in Table 11. All test assets, including industry portfolios (i.e., significant at the 5 percent level), are priced by this aggregate market index. Most of the *t*-values for different test assets exceed the recommended 3.0 threshold. For size-profit-investment portfolios the *t*-value reaches a high of 5.9, which is extraordinary in view of the very low and insignificant *t*-values associated with the market factor in almost all published asset pricing studies. Except for the value-investment portfolios, $R^2$ values range from 41 percent to 76 percent,

which implies relatively high goodness-of-fit for a single factor model. Interestingly, all of the estimated mispricing terms $\hat{\alpha}$ using $I_8$ as the market factor are insignificant. These mispricing results bode well for the single factor CAPM. We infer that multifactor market index $I_8$ provides a superior market index for researchers to employ in asset pricing studies.

**Table 9.** Cross-sectional asset pricing tests of a single industry index incorporating 30 industry factors using relative Sharpe ratio weights for a variety test asset portfolios: July 1963–December 2016. Based on CRSP stock return data in the sample period July 1963 to December 2016, this table provides Fama and MacBeth (1973) cross-sectional tests of a single industry index incorporating 30 industry factors denoted *IND*. Each industry factor is defined as a pure long/short portfolio that is long a particular industry and short an equal quantity of Treasury bills. The single industry index *IND* is formed by combining the 30 industry factors based on the relative Sharpe ratio weighted methods defined in Section 2. Monthly returns for a variety of test assets are downloaded from French's data website: 30 industry portfolios, 25 size-value portfolios, 25 value-investment portfolios, 25 profit-investment portfolios, 25 size-investment portfolios, 25 size-profit portfolios, 32 size-value-investment portfolios, 32 size-value-profit portfolios, and 32 size-profit-investment portfolios (see French's website for details of these portfolios). A time-series regression using monthly excess returns is run for the full sample period to estimate industry betas for each of the tests assets with respect to industry index *IND*. Following the procedure in the text and Table 2, monthly industry index returns are scaled to contain no leverage. Cross-sectional tests are conducted by estimating monthly rolling cross-sectional regressions with excess returns and full sample industry betas for all sample months. The resultant monthly series of estimated market prices of industry beta risk are averaged over all sample months to estimate $\hat{\lambda}_{IND}$ (and associated *t*-statistics are in parentheses).

| Test Assets | $\hat{\alpha}$ | $\hat{\lambda}_{IND}$ | Adj. $R^2$ |
|---|---|---|---|
| 30 industry portfolios | 0.06 | 0.82 | 0.65 |
| | (0.19) | (2.50) | |
| 25 size-value portfolios | 2.08 | −2.23 | 0.29 |
| | (5.26) | (−3.71) | |
| 25 value-investment portfolios | 1.63 | −1.54 | 0.32 |
| | (3.56) | (−2.28) | |
| 25 profit-investment portfolios | 0.47 | 0.19 | −0.04 |
| | (1.24) | (0.32) | |
| 25 size-investment portfolios | 2.47 | −2.85 | 0.57 |
| | (5.46) | (−4.27) | |
| 25 size-profit portfolios | −0.15 | 1.39 | 0.12 |
| | (0.416) | (2.74) | |
| 32 size-value-investment portfolios | 2.45 | −2.79 | 0.49 |
| | (5.11) | (−4.11) | |
| 32 size-value-profit portfolios | 1.92 | −1.94 | 0.07 |
| | (3.98) | (−2.50) | |
| 32 size-profit-investment portfolios | 1.53 | −1.37 | 0.04 |
| | (4.43) | (−2.69) | |

*3.7. Is Momentum a Strong Factor?*

Because momentum was significantly priced after removing the excess returns of different multifactor market indexes from the 25 size-value test assets (see Table 7), we designated it as a possible strong factor. As mentioned earlier, strong factors supplement multifactor market indexes to improve model specification.

In Table 12 we report the cross-sectional tests for two-factor models comprised of multifactor market indexes $I_5$, $I_6$, $I_7$, and $I_8$ augmented with the momentum factor (MOM). Note that $I_6$, $I_7$, and $I_8$ contain the momentum multifactor. Referring to Table 12, we find that momentum loadings are significantly priced in the following tests: Panel A for the 25-size-value test asset portfolios in combination with $I_5$, $I_6$, and $I_7$ but not $I_8$; Panel D for 25 size-investment portfolios in combination with $I_8$; Panel F for 32 size-value-investment portfolios in combination with $I_5$, $I_6$, and $I_7$ but not $I_8$; and Panel H for 32 size-profit-investment portfolios in combination with all four multifactor market indexes.

**Table 10.** Cross-sectional asset pricing tests of a simple industry index incorporating equal-weighted industry factors for a variety test asset portfolios: July 1963–December 2016. Based on CRSP stock return data in the sample period July 1963 to December 2016, this table provides Fama and MacBeth (1973) cross-sectional tests of a simple industry index denoted as $SIND$. Each industry factor is defined as a pure long/short portfolio that is long a particular industry and short an equal quantity of Treasury bills. The single industry index $SIND$ is formed by combining the 30 industry factors into an equal-weighted portfolio. Monthly returns for a variety of test assets are downloaded from French's data website: 25 value-investment portfolios, 25 profit-investment portfolios, 25 size-investment portfolios, 25 size-profit portfolios, 32 size-value-investment portfolios, 32 size-value-profit portfolios, 32 size-profit-investment portfolios, and 30 industry portfolios (see French's website for details of these portfolios). A time-series regression using monthly excess returns is run for the full sample period to estimate industry betas for each of the tests assets with respect to market index $SIND$. Cross-sectional tests are conducted by estimating monthly rolling cross-sectional regressions with excess returns and full sample industry betas for all sample months. The resultant monthly series of estimated market prices of industry beta risk are averaged over all sample months to estimate $\hat{\lambda}_{SIND}$ (and associated $t$-statistics are in parentheses).

| Test Assets | $\hat{\alpha}$ | $\hat{\lambda}_{SIND}$ | Adj. $R^2$ |
|---|---|---|---|
| 30 industry portfolios | 0.64 | −0.03 | −0.03 |
| | (2.98) | (−0.09) | |
| 25 size-value portfolios | 0.89 | −0.16 | −0.03 |
| | (2.27) | (−0.34) | |
| 25 value-investment portfolios | 0.51 | 0.16 | −0.03 |
| | (3.56) | (−2.28) | |
| 25 profit-investment portfolios | 1.21 | −0.67 | 0.16 |
| | (4.45) | (−1.89) | |
| 25 size-investment portfolios | 0.81 | −0.08 | −0.04 |
| | (2.78) | (−0.20) | |
| 25 size-profit portfolios | 0.06 | 0.62 | 0.14 |
| | (0.17) | (1.52) | |
| 32 size-value-investment portfolios | 0.23 | 0.51 | 0.08 |
| | (0.66) | (1.22) | |
| 32 size-value-profit portfolios | 0.63 | 0.08 | −0.03 |
| | (1.83) | (0.20) | |
| 32 size-profit-investment portfolios | 0.67 | 0.02 | −0.03 |
| | (2.23) | (0.05) | |

We interpret these results to suggest that momentum is a possible strong factor for all four multifactor market indexes tested, as it continues to be significantly priced in some test assets even if included in respective multifactor market indexes. Nonetheless, as the efficiency of multifactor market indexes increases via the addition of more feasible factors (e.g., based on cross-sectional $t$-values), the hurdle for strong factors such as momentum will increase. It is conceivable that future research using more efficient multifactor market indexes will eliminate momentum as a possible strong factor.

*3.8. Discussion*

Multifactor market indexes have both academic and practical applications. In academic studies, as discussed in Section 2, many different asset pricing models are popular in the literature nowadays. Certainly more models with innovative factors will be proposed in coming years. Which factors should be used by researchers? Our multifactor market index approach enables researchers to reduce this problem to a manageable set of aggregate indexes and strong factors. In this way, not only can all discovered factors be incorporated into multifactor aggregate indexes, but parsimonious models can be specified for broad usage in academic research. For practitioners, multifactor market indexes represent investable strategies to construct efficient portfolios for investment purposes. Hence, portfolio managers can utilize significant factors in academic studies to boost their returns per unit risk for clients. In turn, Markowitz's (1959) mean-variance portfolio theory can be applied to create well diversified, efficient portfolios.

**Table 11.** Cross-sectional asset pricing tests of a multifactor market index incorporating five popular multifactors and 30 industry factors for a variety test asset portfolios: July 1963–December 2016. Based on CRSP stock return data in the sample period July 1963 to December 2016, this table provides Fama and MacBeth (1973) cross-sectional tests of a multifactor market index denoted $I_8$ incorporating five popular multifactors and 30 industry factors. Each industry factor is defined as a pure long/short portfolio that is long a particular industry and short an equal quantity of Treasury bills. Index $I_8$ is formed by adding five popular multifactors (viz., size, value, profit, investment, and momentum) plus 30 industry factors to the CRSP index based on the relative Sharpe ratio weighted methods defined in Section 2. Monthly returns for a variety of test assets are downloaded from French's data website: 30 industry portfolios, 25 value-investment portfolios, 25 profit-investment portfolios, 25 size-investment portfolios, 25 size-profit portfolios, 32 size-value-investment portfolios, 32 size-value-profit portfolios, and 32 size-profit-investment portfolios (see French's website for details of these portfolios). A time-series regression using monthly excess returns is run for the full sample period to estimate CAPM betas for each of the tests assets with respect to market index $I_8$. Following the procedure in the text and Table 2, monthly market index returns are scaled to contain no leverage. Cross-sectional tests are conducted by estimating monthly rolling cross-sectional regressions with excess returns and full sample betas for all sample months. The resultant monthly series of estimated market prices of beta risk are averaged over all sample months to estimate $\hat{\lambda}_M$ (and associated *t*-statistics are in parentheses).

| Test Assets | $\hat{\alpha}$ | $\hat{\lambda}_M$ | Adj. $R^2$ |
|---|---|---|---|
| 30 industry portfolio | 0.33 | 0.20 | 0.41 |
| | (1.27) | (2.01) | |
| 25 size-value portfolios | −0.20 | 0.59 | 0.63 |
| | (−0.82) | (4.43) | |
| 25 value-investment portfolios | 0.36 | 0.20 | 0.12 |
| | (1.49) | (2.11) | |
| 25 profit-investment portfolios | 0.06 | 0.39 | 0.76 |
| | (0.25) | (4.15) | |
| 25 size-investment portfolios | −0.02 | 0.47 | 0.65 |
| | (−0.08) | (4.02) | |
| 25 size-profit portfolios | 0.12 | 0.38 | 0.79 |
| | (0.56) | (3.51) | |
| 32 size-value-investment portfolios | −0.00 | 0.46 | 0.62 |
| | (−0.02) | (3.82) | |
| 32 size-value-profit portfolios | −0.17 | 0.59 | 0.68 |
| | (−0.71) | (5.03) | |
| 32 size-profit-investment portfolios | −0.07 | 0.49 | 0.71 |
| | (−0.34) | (5.91) | |

**Table 12.** Cross-sectional asset pricing tests of a two-factor model containing different multifactor market indexes augmented with the momentum factor for a variety of test asset portfolios: July 1963–December 2016. Based on CRSP stock return data in the sample period July 1963 to December 2016, this table provides Fama and MacBeth (1973) cross-sectional tests of two-factor models comprised of different multifactor market indexes augmented with the momentum factor. Downloaded from Kenneth French's website, the multifactors are: size (SMB), value (HML), profit (RMW), capital investment (CMA), and momentum (MOM). Multifactor market indexes are formed using the following steps. First, the size (SMB) factor monthly returns are regressed on CRSP index ($I_1$) excess returns over the Treasury bill rate. The residual term from this regression is utilized as the orthogonalized factor. Second, this orthogonalized size factor is added to the CRSP index using Equation (11) to compute the return for the new multifactor market index $I_2$ as $R(I_2) = R(I_1) + x_1$SMB. Third, value (HML) factor returns are regressed on the new $R(I_1) + x_1$SMB portfolio excess returns to obtain the orthogonalized value factor. This residual value factor is added to the CRSP + SMB portfolio to get market index $I_3$ = CRSP + $x_1$SMB + $x_2$HML. Fourth, the last step is repeated to sequentially create market index $I_4$ = CRSP + $x_1$SMB + $x_2$HML + $x_3$RMW, market index $I_5$ = CRSP + $x_1$SMB + $x_2$HML + $x_3$RMW + $x_4$CMA, and market index $I_6$ = CRSP + $x_1$SMB + $x_2$HML + $x_3$RMW + $x_4$CMA + $x_5$MOM. We drop the CMA multifactor to form market index $I_7$ = CRSP + $x_1$SMB + $x_2$HML + $x_3$RMW + $x_5$MOM. In addition, we form market index $I_8$ = CRSP + $x_1$SMB + $x_2$HML + $x_3$RMW + $x_5$MOM + ($x_6$ to $x_{35}$) × (1 to 30) industry factors, which are defined as the industry index return minus the Treasury bill rate. Following the procedure in the text and Table 2, monthly market index returns are scaled to contain no leverage. Monthly returns for a variety of test assets are downloaded from French's data website: 25 value-investment portfolios, 25 profit-investment portfolios, 25 size-investment portfolios, 25 size-profit portfolios, 32 size-value-investment portfolios, 32 size-value-profit portfolios, 32 size-profit-investment portfolios, and 30 industry portfolios (see French's website for details of these portfolios). A time-series regression is run for the full sample period to estimate multifactor market and momentum factors' betas for each of the tests assets. Time-series regression analyses are repeated for the $I_5$, $I_6$, $I_7$, and $I_8$ multifactor market indexes. Cross-sectional tests are conducted by estimating monthly rolling cross-sectional regressions with excess returns and full sample betas for all sample months. The resultant monthly series of estimated market prices of beta risk are averaged over all sample months to estimate $\hat{\lambda}_M$ (and associated *t*-statistics are in parentheses).

**Panel A: 25 Size-Value Portfolios**

| Factors | $\hat{\alpha}$ | *t*-Value | $\hat{\lambda}_M$ | *t*-Value | $\hat{\lambda}_{MOM}$ | *t*-Value | Adj. $R^2$ |
|---|---|---|---|---|---|---|---|
| $I_5$, MOM | 0.74 | 3.43 | 0.25 | 3.82 | 1.87 | 3.42 | 0.65 |
| $I_6$, MOM | 0.74 | 3.43 | 0.50 | 4.50 | 1.87 | 3.42 | 0.65 |
| $I_7$, MOM | 0.37 | 1.43 | 0.73 | 5.43 | 2.47 | 3.93 | 0.79 |
| $I_8$, MOM | −0.27 | −1.02 | 0.48 | 2.65 | 0.55 | 0.66 | 0.64 |

**Panel B: 25 Value-Investment Portfolios**

| Factors | $\hat{\alpha}$ | *t*-Value | $\hat{\lambda}_M$ | *t*-Value | $\hat{\lambda}_{MOM}$ | *t*-Value | Adj. $R^2$ |
|---|---|---|---|---|---|---|---|
| $I_5$, MOM | 0.46 | 2.65 | 0.17 | 2.36 | 0.06 | 0.10 | 0.52 |
| $I_6$, MOM | 0.46 | 2.65 | 0.15 | 1.20 | 0.06 | 0.10 | 0.52 |
| $I_7$, MOM | 0.30 | 1.36 | 0.27 | 1.48 | 0.43 | 0.64 | 0.48 |
| $I_8$, MOM | 0.18 | 0.73 | 0.08 | 0.73 | −0.57 | −1.10 | 0.44 |

**Panel C: 25 Profit-Investment Portfolios**

| Factors | $\hat{\alpha}$ | *t*-Value | $\hat{\lambda}_M$ | *t*-Value | $\hat{\lambda}_{MOM}$ | *t*-Value | Adj. $R^2$ |
|---|---|---|---|---|---|---|---|
| $I_5$, MOM | 0.47 | 2.46 | 0.25 | 3.27 | 0.53 | 1.02 | 0.57 |
| $I_6$, MOM | 0.47 | 2.46 | 0.29 | 2.74 | 0.53 | 1.02 | 0.57 |
| $I_7$, MOM | 0.09 | 0.36 | 0.42 | 2.99 | 0.45 | 0.86 | 0.55 |
| $I_8$, MOM | 0.00 | 0.01 | 0.37 | 3.47 | 0.59 | 1.11 | 0.76 |

**Table 12.** *Cont.*

**Panel D: 25 Size-Investment Portfolios**

| Factors | $\hat{\alpha}$ | *t*-Value | $\hat{\lambda}_M$ | *t*-Value | $\hat{\lambda}_{MOM}$ | *t*-Value | Adj. $R^2$ |
|---|---|---|---|---|---|---|---|
| $I_5$, MOM | 0.51 | 2.94 | 0.29 | 4.49 | 0.49 | 0.74 | 0.62 |
| $I_6$, MOM | 0.51 | 2.94 | 0.32 | 2.76 | 0.49 | 0.74 | 0.62 |
| $I_7$, MOM | 0.16 | 0.76 | 0.48 | 3.82 | 0.82 | 1.43 | 0.62 |
| $I_8$, MOM | 0.03 | 0.14 | 0.57 | 4.02 | 1.66 | 2.65 | 0.65 |

**Panel E: 25 Size-Profit Portfolios**

| Factors | $\hat{\alpha}$ | *t*-Value | $\hat{\lambda}_M$ | *t*-Value | $\hat{\lambda}_{MOM}$ | *t*-Value | Adj. $R^2$ |
|---|---|---|---|---|---|---|---|
| $I_5$, MOM | 0.44 | 2.50 | 0.29 | 3.65 | 0.11 | 0.19 | 0.75 |
| $I_6$, MOM | 0.44 | 2.50 | 0.26 | 2.67 | 0.11 | 0.19 | 0.75 |
| $I_7$, MOM | 0.25 | 1.35 | 0.27 | 2.70 | 0.16 | 0.27 | 0.84 |
| $I_8$, MOM | −0.03 | −0.14 | 0.30 | 3.15 | 0.10 | 0.17 | 0.81 |

**Panel F: 32 Size-Value-Investment Portfolios**

| Factors | $\hat{\alpha}$ | *t*-Value | $\hat{\lambda}_M$ | *t*-Value | $\hat{\lambda}_{MOM}$ | *t*-Value | Adj. $R^2$ |
|---|---|---|---|---|---|---|---|
| $I_5$, MOM | 0.60 | 3.25 | 0.26 | 4.18 | 1.27 | 2.56 | 0.49 |
| $I_6$, MOM | 0.60 | 3.25 | 0.41 | 4.05 | 1.27 | 2.56 | 0.49 |
| $I_7$, MOM | 0.30 | 1.45 | 0.68 | 4.08 | 2.15 | 3.29 | 0.60 |
| $I_8$, MOM | −0.14 | −0.57 | 0.33 | 2.82 | 0.02 | 0.04 | 0.70 |

**Panel G: 32 Size-Value-Profit Portfolios**

| Factors | $\hat{\alpha}$ | *t*-Value | $\hat{\lambda}_M$ | *t*-Value | $\hat{\lambda}_{MOM}$ | *t*-Value | Adj. $R^2$ |
|---|---|---|---|---|---|---|---|
| $I_5$, MOM | 0.46 | 2.26 | 0.26 | 4.56 | 0.31 | 0.53 | 0.66 |
| $I_6$, MOM | 0.46 | 2.27 | 0.27 | 2.56 | 0.31 | 0.53 | 0.66 |
| $I_7$, MOM | 0.20 | 0.85 | 0.38 | 3.42 | 0.57 | 1.04 | 0.74 |
| $I_8$, MOM | −0.34 | −1.26 | 0.35 | 2.91 | −0.26 | −0.44 | 0.77 |

**Panel H: 32 Size-Profit-Investment Portfolios**

| Factors | $\hat{\alpha}$ | *t*-Value | $\hat{\lambda}_M$ | *t*-Value | $\hat{\lambda}_{MOM}$ | *t*-Value | Adj. $R^2$ |
|---|---|---|---|---|---|---|---|
| $I_5$, MOM | 0.54 | 3.02 | 0.34 | 5.54 | 1.36 | 2.85 | 0.76 |
| $I_6$, MOM | 0.54 | 3.02 | 0.50 | 6.20 | 1.36 | 2.85 | 0.76 |
| $I_7$, MOM | 0.21 | 1.09 | 0.59 | 6.96 | 1.51 | 3.30 | 0.76 |
| $I_8$, MOM | −0.04 | −0.18 | 0.51 | 7.01 | 1.24 | 2.81 | 0.71 |

**Panel I: 30 Industry Portfolios**

| Factors | $\hat{\alpha}$ | *t*-Value | $\hat{\lambda}_M$ | *t*-Value | $\hat{\lambda}_{MOM}$ | *t*-Value | Adj. $R^2$ |
|---|---|---|---|---|---|---|---|
| $I_5$, MOM | 0.65 | 3.79 | 0.02 | 0.28 | 0.32 | 0.65 | −0.01 |
| $I_6$, MOM | 0.65 | 3.79 | 0.06 | 0.61 | 0.32 | 0.65 | −0.01 |
| $I_7$, MOM | 0.60 | 3.36 | 0.11 | 0.96 | 0.43 | 0.87 | 0.03 |
| $I_8$, MOM | 0.30 | 1.34 | 0.20 | 1.98 | 0.37 | 0.84 | 0.39 |

## 4. Conclusions

This paper sought to mitigate the model mall problem associated with the growing list of factors in asset pricing. To do this, we proposed the construction of efficient multifactor market indexes that combine the CRSP index with popular investable multifactors. To demonstrate this approach, size, value, profit, capital investment, and momentum multifactors were sequentially added to the CRSP index. In the sample period July 1963 to December 2016, as multifactors were added to the CRSP index, multimarket market indexes became increasingly less correlated with the CRSP index and more efficient than the CRSP index. Importantly, market betas associated with most of these multifactor market indexes were significantly priced in cross-sectional asset pricing tests with economically meaningful market prices of risk. With the exception of industry portfolio test assets, the *t*-values associated with multifactor market index betas generally exceeded the recommended 3.0 threshold for the significance of an asset pricing factor. In addition, we found that multifactors were less likely to be significantly priced in cross-sectional tests when the market index contained the respective multifactors. One exception was the momentum factor, which was significantly priced even when contained in multifactor market indexes.

We inferred that momentum may be a strong factor in the sense that it can serve to augment multifactor market indexes in models comprised of two or more factors.

Further analyses demonstrated how a single industry index can be constructed by combining numerous industry indexes. This new aggregate industry index was significantly priced among industry portfolios, for which the CRSP index and multifactors are typically not priced. When popular multifactors and industry factors were combined with the CRSP index into a multifactor market index, the resultant market factor was significantly priced across a wide variety of test assets including industry portfolios. Exemplary of our findings, a very high $t$-value of 5.91 was achieved for size-value-investment portfolios. In tests of this market index across different test assets, the goodness-of-fit was relatively high and comparable to traditional multifactor models, and no mispricing was detected.

When multifactor market indexes were augmented with the momentum factor in a two-factor model, momentum loadings were significant in a number of different test assets, but not all test assets. We conclude that momentum is a possible strong factor that is priced at times even when contained in the two-factor model's multifactor market index. Future research is needed to determine if, as multifactor market indexes become more efficient with the inclusion of more feasible factors, momentum remains a priced factor.

Based on our empirical results, we conclude that: (1) multifactor market indexes are more efficient than the CRSP market index, and (2) market beta is significantly priced for these multivariate market index proxies. Hence, we infer that multifactors are not separate risk factors but rather help jointly in combination with a general stock market index to better proxy efficient market portfolios. An important implication of our findings is that asset pricing models incorporating a market factor (including unconditional and conditional empirical models) would benefit from employing multifactor market indexes rather than the CRSP index or other general market indexes. More parsimonious low-dimensional models can be developed that incorporate multifactor market indexes to substantially reduce factor and model selection problems. That is, rather than creating an increasing number of asset pricing models featuring different multifactors, most of the multifactors can be productively utilized to form more efficient multifactor market indexes. Most investable factors will likely be absorbed into a multifactor market index. Noninvestable factors, such as market volatility, macroeconomic state variables, etc., cannot be combined into an investable multifactor market index and, therefore, could be prospects as strong factors in a low-dimensional model. More generally, by combining many proven factors into a single aggregate market index, it is possible that better asset pricing models can be constructed that capture the collective significance of numerous risk dimensions associated with a wide array of factors.

Future research is recommended on efficient multifactor market indexes to reduce the problems of factor and model selection in various asset pricing applications, such as event studies of corporate and other major events, investment performance studies of mutual funds and hedge funds, and other studies that rely upon asset pricing models in their empirical analyses. Importantly, a practical implication is that investors can readily form multifactor market indexes by combining (for example) a tradeable S&P 500 index with ETFs related to size, value, momentum, and other multifactors. These new aggregate indexes have the potential to outperform the S&P 500 index, a common benchmark used to evaluate institutional investors. Investors seeking portfolios with higher returns per unit risk would benefit from efficient ETFs and other multifactor index products, especially those saving for retirement.

**Author Contributions:** W.L. worked on the empirical results; J.W.K. wrote the text. All authors have read and agreed to the published version of the manuscript.

**Funding:** This research received no external funding.

**Data Availability Statement:** All data is available from Kenneth French's website.

**Conflicts of Interest:** The authors declare no conflict of interest.

## Notes

1.  See also Lettau and Pelger (2020), who utilized Principal Component Analysis (PCA) methods to reduce the number of potential factors to a parsimonious set of latent factors. They found that five factors with economic content help to explain the cross-section and time-series of returns.

2.  For example, the three moment CAPM (Rubinstein (1973) and Kraus and Litzenberger (1976)), consumption CAPM (Breeden 1979), conditional CAPM (Jagannathan and Wang (1996) and Ferson and Harvey (1999)), liquidity-based models (Pastor and Stambaugh (2003), Acharya and Pedersen (2005), and Li et al. (2019)), intertemporal CAPM (ICAPM) (Petkova 2006), interest-rate-based models Campbell (1996), cross-factor models Fama and French (2020), among others.

3.  As such, the intercept equals zero, and the beta factor loading matrix is unchanged.

4.  Numerous papers have found that various long-only market indexes are not mean-variance efficient portfolios (e.g., see Gibbons et al. (1989); Gibbons (1982); Jobson and Korkie (1982); Kandel (1984); Shanken (1985, 1986); Kandel and Stambaugh (1987a, 1987b); Gibbons et al. (1989); Haugen and Baker (1991); MacKinlay and Richardson (1991); Zhou (1993); Brière et al. (2013), and others). According to Brennan and Lo (2010), it is virtually impossible for long-only market indexes to be efficient portfolios. Supporting this proposition, many studies have found that short positions are needed to achieve efficiency (e.g., see Pulley (1981); Levy (1983); Kallberg and Ziemba (1983); Kroll et al. (1984); Green and Hollifield (1992); Jagannathan and Ma (2003); Brennan and Lo (2010); Levy and Ritov (2010), and others). More generally, Kothari et al. (1995) have argued that the equity portfolio most highly correlated with the market portfolio is efficient.

5.  Also, if the expected return of the zero-cost factor portfolio is zero, the Sharpe ratio will not be increased by adding it to a candidate market index. Note that any zero-cost asset that is uncorrelated with the tangency portfolio has an expected return equal to the riskless rate such that its excess return is zero.

6.  As shown there, the definitions of the multifactors are as follows: size (SMB) is the average return on the nine small stock portfolios minus the average return on the nine big stock portfolios; value (HML) is the average return on the two value portfolios minus the average return on the two growth portfolios; profit (RMW) is the average return on the two robust operating profitability portfolios minus the average return on the two weak operating profitability portfolios; capital investment (CMA) is the average return on the two conservative investment portfolios minus the average return on the two aggressive investment portfolios; and momentum (MOM) is the average return on the two high prior return portfolios minus the average return on the two low prior return portfolios.

7.  These weights are averages using different orders of entry in forming indexes based on a rotation of factors. For example, $I_3$ can be formed by starting with CRSP, adding SMB, and then adding HML. Alternatively, we can start with SMB, add HML, and then CRSP. Finally, we could add them in the order HML, SMB, and CRSP. We average weights for each factor across these rotational combinations of order entry. While the order in which portfolios is combined changes their relative weights, the Sharpe ratios and other performance metrics of respective aggregate indexes as well as forthcoming cross-sectional tests were little changed.

8.  Rearranging terms, we have: $R(I_{2,t}) = R(I_{1,t}) + x_1 F_{1,t}^{new} = R(I_{1,t}) + x_1[F_{1,t} - \beta_1 R(I_{1,t})]$. Since both $R(I_{1,t})$ and $F_{1,t}$ are deleveraged portfolios, we can deleverage $R(I_{2,t})$ by dividing its return by $1 + x_1(1 - \beta_1)$.

9.  See also Roll (1977, p. 130), who also observed that most proxies for the market portfolio are very highly correlated.

10. They also included a time-varying market factor to capture beta instability over time (i.e., the BAA minus AAA bond yield spread), which was found to be significantly priced. Hence, they concluded that, even though the static CAPM assuming constant beta over time is not supported, the conditional CAPM allowing betas and expected returns to vary over time is supported.

11. The GRS test statistic has a noncentral $F$ distribution with degrees of freedom $N$ (25 portfolios) and $T - N - 1$ ($T$ 654 months). As the noncentrality parameter increases, the probability of rejecting a false null hypothesis tends to increase. According to tests in Gibbons et al. (1989, pp. 1130–38), the power of our tests should be sufficient to detect deviations from the efficiency of the index.

12. Following standard practice, adjusted $R^2$ values are estimated by regressing the average excess returns for test asset portfolios in the full sample period on their full sample beta estimates.

13. By contrast, CMA is positive but not significantly priced for market indexes $I_1$ and $I_2$ based on the CRSP index and CRSP + SML market index, respectively, but it is significantly priced with market indexes $I_3$ and $I_5$ containing the CMA portfolio. These unexpected results for CMA are difficult to interpret due to being negatively (rather than positively) priced when significant.

14. Multifactor market index $I_7$ does not eliminate the significance of the SMB, HML, RMW, and MOM factors even though these factors are contained in this index. However, the markedly lower $R^2$ value of 42 percent for this market index indicates that these factors' residual explanatory power is substantially diminished by market index $I_7$.

15. Early work on industry as an asset pricing factor by King (1966) and Meyers (1973) found that, based on principal components analyses of U.S. stock returns, most components could not be identified with specific industry groups. Components were only weakly associated with industry classifications.

16. Unreported in Table 9, we also tested a two-factor model with the excess return on the CRSP index ($I_1$) and industry factor (IND). Using 30 industry portfolios as test assets, both factors are significantly priced with $t$-values of 2.12 and 2.98, respectively. The correlation between these two factors was relatively high at 0.60.

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
