# Peer review of "Multifactor Market Indexes"

_jrfm, doi:10.3390/jrfm15040155_

Round 1

Reviewer 1 Report

I am pleased to have the opportunity to review this research paper. This study attempted to explore a Multifactor Market Indexes. Although the topic of this research study is interesting and fits within the journal scope, I think authors should apply the comments indicated below to increase the quality of research justification, contributions and findings. The manuscript know lacks in scientific style and structure.

First of all, paper research gap. Please improve this part in introduction section. Introduction is very general and lacked alignment to the research findings, no discussion was provided to derive the implication from. Theoretical and pragmatics implication are vague and need to be better aligned with this paper theoretical underpinnings and proposed process. Furthermore, there is insufficient support and weak arguments in support of the objective that is proposed as well as the model developed. In the final part of the introduction the objectives proposed, originality and gap that would be better covered. Also how the author will perform the methodology.

the topic of this research study is interesting and fits within the journal scope, I think authors should apply the comments indicated to increase the quality of research justification, contributions and findings

What is the originality of this research?  Paper research gap and originality should be better presented at the end of introduction section

Please consider this structure for manuscript final part.

-Discussion

-Conclusion

-Managerial Implication

-Practical/Social Implications

-Discussion needs to be a coherent and cohesive set of arguments that take us beyond this study in particular, and help us see the relevance of what authors have proposed. Authors should create an independent “Discussion” section. Author need to contextualize the findings in the literature, and need to be explicit about the added value of your study towards that literature. Also other studies should be cited to increase the theoretical background of each of the method used. Findings should be contextualized in the literature and should be explicit about the added value of the study towards the literature. Limitations and future research

Questions to be answered:

What practical/professional and academic consequences will this study have for the future of scientific literature (theoretical contributions)?

Why is this study necessary? should make clear arguments to explain what is the originality and value of the proposed model. This should be stated in the final paragraphs of introduction and conclusion sections.

Author Response

Comments and Responses to Reviewer 1:

  1. Need to improve the Introduction section.  Discussion is very general.  At the end of the Introduction section, the paper research gap and originality should be better presented.

Response: We have re-written the Introduction.  Some of the material is moved to Section 2 to better focus the motivation for the present study.  Additionally, we more clearly state the problem of multiple acceptable factors in the literature.  Our solution is to combine them into a single aggregate factor.  Empirical tests show that this single factor is efficient and highly significant in cross-sectional asset pricing tests.  Also, we discuss the construction of an industry index as a new factor in the literature.  Finally, we introduce the concept of a strong factor that is priced even if included into an aggregate index factor.  Thus, our paper makes a number of contributions to the multifactor model literature.

We did have discussion of the research contributions of the paper at the end of the Introduction section:

“Multifactor market indexes that are significantly priced can potentially lead to more parsimonious and robust asset pricing models. Since the market factor is commonly used in many areas of corporate, investment, and institutional finance, widespread applications of multifactor market indexes are possible in future studies. Another implication is that similar efficient aggregate indexes can be constructed in the real world for investment purposes, thereby benefiting many investors including those saving for retirement.”

These are original contributions that help to resolve problems in specifying asset pricing models with numerous available factors.

  1. An independent discussion section should be created and discuss added value of the study to the literature.

Response: We have added a Discussion section just before the Conclusion section to discuss academic and practitioner uses of multifactor market indexes.  As recommended, the final paragraphs of the Introduction and Conclusion sections review these contributions also.

Reviewer 2 Report

This article sifts through the factor zoo and orthogonalizes additional factors in order to judge their statistical significance. Multiple factors models are tested via the GRS test (1989,ECTA). The authors present a reduced set of factors based on a Fama-Macbeth cross-sectional regression tests.

I have a few comments and suggestions for the authors to consider:

1) The coefficients $\beta_{k+1}$ in Equation (10) and $x_{k+1}$ in Equation (11) are estimated with error and perhaps we need to consider these estimation errors into the construction of the orthogonalized multifactor models (equations on top of page 11).

2) The GRS test also differs considerably in power depending on the sample sizes (T) versus the cross-section (N). Perhaps the authors could mention what the power of the test is assuming a typical non-centrality parameter value.

3) Finally, I find the proposed methodology quite reminiscent of the mean-variance spanning literature [Huberman and Kandel (1987,JF) and more recently Kan and Zhou (2012,AEF)]; I think readers might benefit if the authors link their line of work with the mean-variance spanning literature.

Author Response

Comments and Responses to Reviewer 2:

  1. The coefficients in equations (10) and (11) are estimated with error. These errors need to be considered in the construction of multifactor models.

Response:  Recognizing this comment, we added the following text in the paragraph after these equations:  “As noted by an anonymous referee, errors in estimated coefficients will affect these weights and, in turn, multifactor market index returns. This potential bias is beyond the scope of the present research and, therefore, left for future research.”

  1. The GRS test differs in power depending on the sample sizes (t) versus the cross-section (N). Perhaps the author could mention what the power of the test is.

Response:  To address this comment, we added discussion about the power of the GRS test in footnote 12:

“The GRS test statistic has a noncentral $F$ distribution with degrees of freedom $N$ (25 portfolios) and $T-N-1$ ($T$ 654 months). As the noncentrality parameter increases, the probability of rejecting a false null hypothesis tends to increase. According to tests in Gibbons, Ross, and Shanken (1989, pp. 1130-1138), the power of our tests should be sufficient to detect deviations from the efficiency of the index.”

  1. The proposed methodology is reminiscent of mean-variance spanning in the literature by some authors in the literature. The paper should be linked to this literature.

Response:  We have added the references that you cited to the paper:

Huberman and Kandel (1987) and Kan and Zhou (2012).

In the Introduction section, we mention these papers and link them to our paper:

“While numerous authors have proven in theory that a combination of factor-mimicking portfolios is minimum variance efficient (e.g., Huberman and Kandel (1987), Shanken (1987), Fama (1996), Shanken and Weinstein (2006), Kan and Zhou (2008), and others), no previous studies to the authors' knowledge have verified this result in empirical tests. Filling this gap in the literature, we incrementally combine popular factors with the CRSP index to show that the resultant aggregate market indexes are increasingly efficient and priced in the cross section of average stock returns.”

Round 2

Reviewer 1 Report

the work is now much better, I recommend adding more literature that supports the need for the study, before being accepted